


# Validated probabilistic approach to estimate flood direct impacts on the population and assets on European coastlines

Enrico Duo[1,2], Juan Montes[1,3*], Marine Le Gal[1,2], Tomás Fernández-Montblanc[3], Paolo Ciavola[1,2] and Clara Armaroli[4]

5  [1]Department of Physics and Earth Sciences, University of Ferrara, Ferrara, Italy
[2]Consorzio Futuro in Ricerca, Ferrara, Italy
[3]Earth Sciences Department, University of Cadiz INMAR, Avda. República Saharaui s/n, Puerto Real, 11510 Cadiz, Spain
[4]Department of Biological, Geological, and Environmental Sciences, University of Bologna Alma Mater Studiorum, Bologna, Italy

10  *Correspondence to: Juan Montes (juan.montes@uca.es)

**Abstract.** This work presents the approach used to estimate coastal flood impact, developed within the EU H2020 European Coastal Flood Awareness System (ECFAS) Project, for assessing flood direct impacts on population, buildings, and roads along the European coasts. The methodology integrates object-based and probabilistic evaluations to provide uncertainty estimates for damage assessment. The approach underwent a user-driven co-evaluation process, it was applied to 16 test cases across Europe and validated against reported impact data in three major reference cases: Xynthia at La Faute-sur-Mer (France) in 2010, Xaver at Norfolk (UK) in 2013, and Emma at Cadiz (Spain) in 2018. A comparison with grid-based damage evaluation methods was also conducted. The findings demonstrate that the ECFAS Impact approach offers valuable estimates for affected populations, reliable damage assessments for buildings and roads, and improved accuracy compared to traditional grid-based approaches. The methodology also provides information for prevention and preparedness activities, facilitates further evaluations of risk scenarios and cost-benefit analysis of disaster risk reduction strategies. The approach is a tool suitable for large-scale coastal flood impact assessments, offering improved accuracy and operational capability for coastal flood forecasts. It represents a potential advancement of the existing EU-scale impact method used by the European Flood Awareness System (EFAS) for riverine flood warnings. The integration of object-based and probabilistic evaluations, along with uncertainty estimation, enhances the understanding and management of flood impacts along the European coasts.

## 1 Introduction

The assessment of flood impacts is crucial for coastal management, providing insights on consequences of coastal extremes for risk management (e.g., Van Dongeren et al., 2018). Historical loss analysis and scenario-based variations support strategy evaluation (e.g., Sanuy et al., 2018) and participatory risk management (Barquet and Cumiskey, 2018), as required by the Floods Directive (2007/60/EC). Estimations of the impact of forecasted floods could support civil protection actions (Dottori et al., 2017). With climate change and increasing human pressure, flood impacts will likely intensify (European Commission, 2017; Vousdoukas et al., 2018b), and there is a need for accurate predictions at various spatial and temporal scales.





This study focuses on flood direct impacts, which result from physical contact between water and objects, causing immediate and local effects. In contrast, indirect impacts have long-term consequences, affecting local, regional, and larger scales through chain-reaction mechanisms. Assessing indirect impacts is challenging due to their diverse nature and complex processes across

multiple sectors and scales (Meyer et al., 2013; Armaroli et al., 2019). Instead, there are numerous methods to assess direct impacts (Gerl et al., 2016). Because of the heterogeneity of these methods, ongoing investigations are looking into their limitations for appropriate applications (Molinari et al., 2020; Marvi, 2020; Aribisala et al., 2022).

Methods to calculate flood direct impacts primarily focus on population, buildings, and transport networks, which are the most significant exposed elements (Thomas et al., 2019; Marvi, 2020; Koks et al., 2022). For the population, the impact assessment

quantifies the affected individuals, and when data on the characteristics of the exposed population are available (e.g., age, socio-economic status, governance, accessibility), comprehensive risk-based estimates can be derived using social science approaches. Direct impacts on buildings and roads are often measured by the number of affected assets, damage, or financial loss.

Traditional large-scale direct impact assessments often rely on grid-based, meso-scale evaluations, which are known to

overestimate impacts (Molinari et al., 2020). Correcting these evaluations introduces additional uncertainties due to approximations and assumptions. However, object-based evaluations using detailed vector data and vulnerability models offer more accurate damage assessments (Molinari et al., 2020; Aribisala et al., 2022). While typically used for local scale assessments, a few studies (e.g., Van Ginkel et al., 2021) provide valuable insights for novel large-scale applications.

The European Flood Awareness System (EFAS; https://www.efas.eu/en) represents the current European-scale operational

application for riverine flood impact assessment. EFAS uses deterministic meso-scale methods to assess impacts on population (affected people, exposure only), infrastructure (affected roads, exposure only), and urban, built-up, and agricultural areas (affected areas identified using land cover data). The method provides evaluations of direct economic losses (Dottori et al., 2017). While the approach has limitations related to dataset approximation and deterministic impact assessment methods, it has demonstrated to produce reliable results considering the continental scale of the application. When evaluating impact

assessment models, it is important to consider the scale of analysis, the magnitude and the uncertainty of the estimations.

Uncertainty evaluation is crucial in impact assessments, influencing disaster prevention, management, and policymaking. Researchers are investigating various sources of uncertainty, with consensus on the main driving factors (Hinkel et al., 2021). Besides the inundation model's performance (hazard component), which determines asset flooding (Vousdoukas et al., 2018a), socio-economic components significantly contribute to uncertainties (de Moel and Aerts, 2011; Jongman et al., 2012;

Figueiredo and Martina, 2016; Nguyen et al., 2016), particularly related to exposure and vulnerability. These effects become more pronounced in the analysis of future scenarios.

Two types of uncertainty affect modelled flood impacts: aleatory and epistemic (Merz and Thieken, 2009; Wagenaar et al., 2016). Aleatory uncertainty arises from choices made in representing variables and processes in the model, such as using a single vulnerability model for all residential buildings without accounting for variability within the category (e.g., detached or

semi-detached). It dominates for small flood events or local domains due to the limited sample size of affected assets. Epistemic



uncertainty stems from incomplete understanding of the system and it is the prevalent uncertainty for the analysis of the effects of large flood events or when applying impact methods on large domains.

Probabilistic modelling is used to address uncertainties in coastal flood impact assessment. These models incorporate evaluations of uncertainty, often expressed as percentile-based ranges, to account for specific sources of uncertainty. For instance, multi-model ensembles introduce uncertainty due to the variability of impact models (Figueiredo et al., 2018). Similarly, applications employing one model with multiple parametrizations or resampling of input data produce outcomes with the uncertainty linked to the variability of input data.

There is a growing interest in applying local object-based approaches to assess coastal flood impacts at large scales and incorporating uncertainty evaluation. This is now feasible due to improved computational capabilities and the availability of comprehensive datasets like the Copernicus Coastal Zone layer (https://land.copernicus.eu/local/coastal-zones), which provides up-to-date Land Cover/Land Use (LC/LU) information for coastal areas. The layer was implemented by the European Environment Agency (EEA) in the framework on the thematic mapping of the Copernicus Land Monitoring Service (CLMS). Another valuable resource is the Open Street Map (OSM) dataset, which offers free object-based vector data.

This paper presents a coastal flood impact assessment approach for estimating direct impacts on population, buildings and roads across Europe. The approach integrates methodologies that prioritize object-based and probabilistic evaluations to provide uncertainty estimates for damage assessment. Developed within the EU H2020 European Copernicus Coastal Flood Awareness System (ECFAS) Project (Grant Agreement No 101004211; www.ecfas.eu), the approach underwent a user-driven co-evaluation process (Velegrakis et al., 2022). Referred to as the ECFAS Impact approach, it was applied to 16 test cases along different European coasts and it was validated against reported impact data in three major reference cases. In this work, a comparison with a grid-based damage evaluation of buildings and roads was also conducted on all test cases.

The ECFAS Impact approach was used in the ECFAS project to generate impact layers for the ECFAS Flood Catalogue (Le Gal et al., 2023b). These flood maps cover over 95% of the European coast and include various scenarios of water levels and storm durations. The impact layers were compiled into the ECFAS Pan-EU Impact Catalogue (Duo et al., 2022). These catalogues were used to implement a proof of concept for a European coastal flood early warning system, which provides notifications based on affected population aggregated by NUTS3 administrative units and is developed considering the EFAS-CEMS framework.

The paper is organized by a detailed description of the test cases, reference cases and the data used to apply and validate the ECFAS Impact approach (Section 2), a detailed description of the approach to evaluate direct impacts on population and assets (Section 3), an overview of the impacts for the reference cases and test cases (Section 4), a comprehensive discussion on the validation with reported impacts, a comparison with grid-based damage evaluations and the limitations of the approach (Section 5) and the conclusions (Section 6).





## 2 Data

### 2.1 Test cases and reference cases

A total of 16 test cases (Table 1) were selected to apply the ECFAS Impact approach. The test cases include 10 extreme events

covering the period 2010-2020, generating considerable flooding and impacts along 15 European coastal sites (Figure 1). These were selected from the database of extreme events and test cases produced in the framework of the ECFAS project (Souto Ceccon et al., 2021). The database, which includes a list of sources of information for each identified test case, was produced based on extensive research of publicly available resources, of information included in the list of activations of the Copernicus Emergency Management Service (CEMS) and other national and European databases. The area of interest (AoI)

of each site (Figure 1) was defined based on the reported affected areas or the AoIs defined for the CEMS activations. Note that all the analyses reported in this work are limited to the AoIs.

Three reference cases were selected from the previous list to implement detailed comparison with reported damages for validation purposes: Xynthia at La Faute-sur-Mer (France) in 2010, Xaver at Norfolk (UK) in 2013, and Emma at Cadiz (Spain) in 2018. General information on the event of the reference cases is summarized in Table 2. The reference cases were

selected because of their dimension in terms of hazard and impacts. The flood extension and water depths used for the impact assessment in the test cases was modelled using the LISFLOOD-FP model (Le Gal et al. 2023a; 2023b) as described in the following section.

**Table 1: Overview of the test cases. The selected reference cases are highlighted in bold.**

| Site Name | Country | Storm Name | Reference Date | AoI Area [km²] | AoI simulated flooded area [km²] | AoI simulated water depth [m] |
|---|---|---|---|---|---|---|
| | | | | | Source: see Section 2.2 | |
| La Baule | France | No name | 02/01/2014 | 60.8 | 6.9 | 0.13 - 2.88 |
| **La Faute-sur-Mer** | **France** | **Xynthia** | **27/02/2010** | **321.6** | **176.4** | **0.15 - 3.54** |
| Lorient | France | No name | 02/01/2014 | 48.0 | 6.0 | 0.12 - 2.82 |
| Warnemunde | Germany | Axel | 05/01/2017 | 7.8 | 0.2 | 0.10 - 0.90 |





| | | | | | | |
|---|---|---|---|---|---|---|
| Wismar | Germany | Axel | 05/01/2017 | 33.9 | 1.0 | 0.11 - 1.36 |
| Laganas | Greece | Ianos | 18/09/2020 | 4.7 | 0.1 | 0.12 - 0.23 |
| Lido delle Nazioni | Italy | Saint Agatha | 05/02/2015 | 81.0 | 44.7 | 0.14 - 4.22 |
| Lido delle Nazioni | Italy | Vaia | 29/10/2018 | 81.0 | 34.9 | 0.15 - 3.89 |
| Lido delle Nazioni | Italy | Detlef | 11/11/2019 | 81.0 | 44.8 | 0.15 - 4.16 |
| Rimini | Italy | Saint Agatha | 05/02/2015 | 148.8 | 5.4 | 0.11 - 1.18 |
| Świnoujście | Poland | Axel | 05/01/2017 | 52.7 | 10.1 | 0.12 - 1.33 |
| Castellon | Spain | Gloria | 20/01/2020 | 3.4 | 0.2 | 0.11 - 0.63 |
| Ebro | Spain | Gloria | 20/01/2020 | 19.5 | 17.6 | 0.34 - 2.83 |
| Girona | Spain | Gloria | 20/01/2020 | 13.2 | 0.7 | 0.11 - 0.96 |
| **Norfolk** | **United Kingdom** | **Xaver** | **06/12/2013** | **207.1** | **52.3** | **0.17 - 3.39** |
| **Cadiz** | **Spain** | **Emma** | **01/03/2018** | **23.9** | **14.7** | **0.15 - 2.51** |





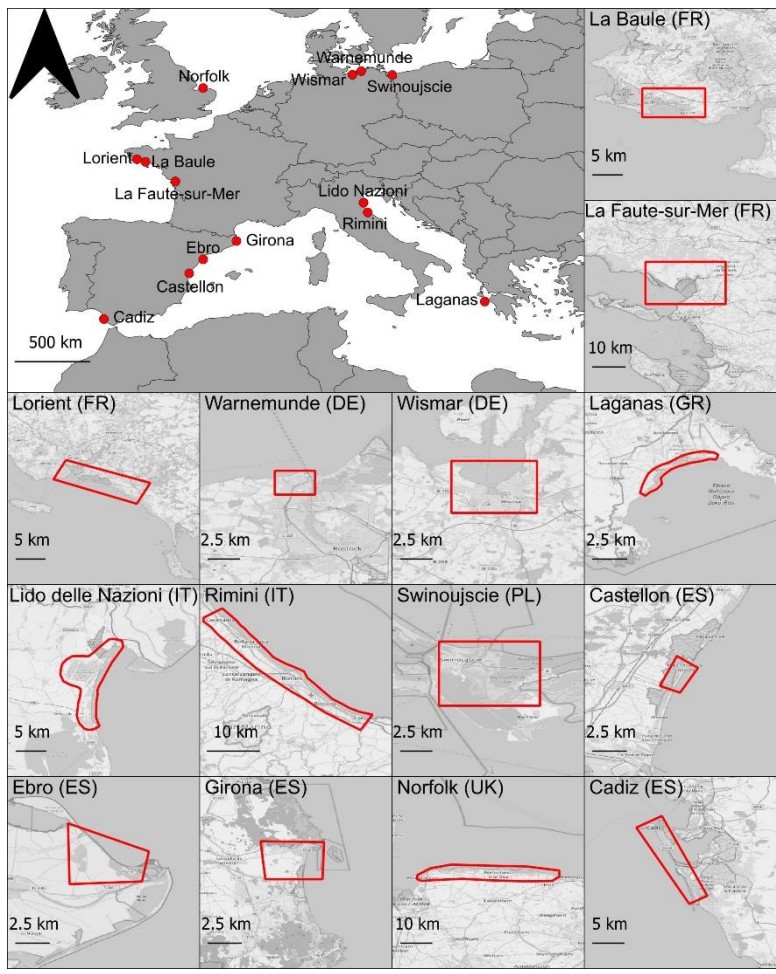

**Figure 1: Overview of the location of the sites of the selected test cases, and AoIs (red polygons).**

**Table 2: Overview of the characteristics of the reference cases events.**

| Reference case event | Dates | Offshore conditions | Consequences | References |
|---|---|---|---|---|
| Xynthia (France) | 27-28 Feb. 2010 | Water Levels: 4.7 m | Affected coast: 200 km<br>Flooded area: 500 km$^2$<br>47 deaths<br>Defence overtopping | Vinet et al., 2012<br>Creach et al., 2015<br>Kolen et al., 2013 |
| Xaver (United | 4-6 Dec. 2013 | Sign. Wave Height: 3.8 m | Flooding of cities, harbours, private | Spencer et al., 2014<br>Spencer et al., 2015 |



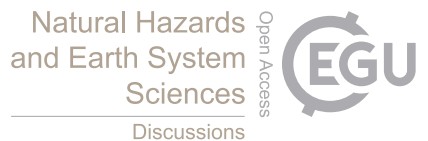

| Kingdom) | | 60-years return level surge | properties, commercial activities, transport infrastructures<br>Cliff collapse<br>Beach Erosion | |
| --- | --- | --- | --- | --- |
| Emma (Spain) | 28 Feb. - 3 Mar. 2018 | Sign. Wave Height: 6.9 m<br>Water Levels: 2.1 m | Flooding of roads, promenades, private and commercial properties<br>Beach Erosion<br>Overwash | Ferreira et al., 2019<br>Plomaritis et al., 2019<br>Talavera et al., 2020<br>Malvarez et al., 2021<br>Montes et al., 2018 |


## 2.2 Flood maps

The flood maps used to represent the coastal flood for the test cases were retrieved from the database of flood maps produced by Le Gal et al. (2023a; 2023b) using the LISFLOOD-FP model (Bates et al., 2005) in the framework of the ECFAS project. The approach utilised a 10 m DEM (COP-DEM-EEA10; European Space Agency and Airbus, 2022) to generate a 100 m

125 resolution grid. The bottom friction was spatially adapted by using literature-based Manning coefficients adjusted to the LC/LU from the Copernicus Coastal Zone layer (see Section 2.3). The flood models were forced with total water level timeseries built by linear addition of the mean sea level, tide, and storm surge components retrieved from Copernicus Marine Environment Monitoring Service (CMEMS) ocean models (for tides, the FES2014 model was used when data was missing; more details in Irazoqui Apecechea et al., 2023), and empirical estimate of the wave set-up based on CMEMS wave models data. More details

130 can be found in Le Gal et al. (2023a; 2023b).

## 2.3 Main datasets for impact assessment

All datasets used to implement the flood impact assessment were collected in the framework of the ECFAS project (Ieronymidi and Grigoriadis, 2022). The Global Human Settlement Population Grid (GHS-POP R2019A; Schiavina et al., 2019) and the ENhancing ACTivity and population mapping 2011 Population Grid (ENACT-POP R2020A; Schiavina et al., 2020), that

135 provide raster-based information about the distribution of population, were used to assess the number of people affected by the flood (see Section 3.2). The GHS-POP includes a static layer of the distribution of people in 2015; the ENACT includes 24 layers describing the population distribution by night and day for each month of the reference year 2011. The OSM vector dataset, that includes information about buildings and transport networks, was used to assess the flood damage to buildings (Section 3.3) and roads (Section 3.4). Note that the OSM coverage of the roads at the EU level is reliable for large-scale

140 evaluations, with almost complete coverage for the EU countries (Barrington-Leigh and Millard-Ball, 2017; Van Ginkel et al.,



2021). However, for buildings, the spatial coverage depends on the country. The Copernicus Coastal Zone (CCZ; https://land.copernicus.eu/local/coastal-zones; Innerbichler et al., 2021) vector layer, that represents the most detailed up-to-date Land Cover / Land Use (LC/LU) layer for coastal areas in Europe, was used in support of the damage assessment for buildings and of the grid-based damage evaluations implemented for comparison purposes (Section 3.6). It represents a highly

detailed dataset compared to CORINE (Büttner et al., 2014) or LUISA (Rosina et al., 2018) for the coastal areas. A detailed overview of the characteristics of the datasets and links to the sources can be found in Appendix A.

## 2.4 Sources of reported impacts

Reported impacts are essential for evaluating the performance of impact models. However, this type of information is often scarce, with qualitative information being more often available than quantitative ones. Databases of micro-scale flood damages

are quite common, but they often represent very local datasets, which are difficult to retrieve, are usually reported in local languages and even more difficult to be used for large-scale analysis. On the other hand, aggregated information on impacts and damages are generally available at different spatial and temporal scales, but they can rarely be used in direct comparisons with simulated impacts, as often data disaggregation and manipulation are needed for comparisons. Additionally, reported damages are often incomplete and reliable estimates might not be available for years after the event (Thieken et al., 2016).

Information on reported impacts were collected and used as ground truth for validation purpose. The data were extracted, georeferenced and characterized by analysing the sources of information included in the database of extreme events and test cases produced in the framework of the ECFAS project (Souto Ceccon et al., 2021). The sources of information include institutional websites, scientific articles, databases, news, technical reports, blogs and videos, among others. The collected information was analysed to build a database of impact markers that the events generated within the AoI of the affected sites.

Impacts were categorised according to the type of impact as defined by the RISC-KIT project (Viavattene et al., 2015), thus discriminating between impacts to the population, buildings/private properties, infrastructures, economy, environment and cultural heritage. Quality indexes were assigned to the identified markers to ensure the control of the reliability of the information using an approach adapted from Sancho-García et al. (2021). This approach employs 3-level indexing of the quality of the spatial and temporal references, and for the level of detail of the information contained in the original source.

For each identified impact marker, when available, the reported local damage in euros was provided. Any additional information that could possibly support the analysis was included for each identified marker.

## 3 Methods

### 3.1 General aspects

The ECFAS Impact Approach integrates methodologies to assess direct impacts on population and assets. Developed

specifically to be applied at the EU scale, object-based, micro-scale methods were preferred when possible. Exposure and vulnerability aspects were considered depending on data availability and reliability. Exposure-based evaluations were preferred





when vulnerability data was not available, or when the assumptions related to the application of vulnerability models generated biased, or very uncertain, results. For population, a grid-based approach was used, while buildings and roads were assessed through object-based methods, incorporating category-based vulnerability.

Impacts were calculated for each affected cell (population), or asset (buildings and roads) based on multiple input data or model ensembles. Probabilistic impacts were generated by resampling an empirical cumulative distribution function (ECDF), generating 1000 scenarios. The total impact in the flooded area was calculated by summing the contribution of each cell/asset for each scenario. The distribution of impact was represented by percentiles (2.5, 50 and 97.5). These evaluations can be repeated to calculate disaggregated impacts by category of asset.

Damages were based on average 2020 prices of the former EU-28, adjusted using Eurostat Real Gross Domestic Product (GDP) statistics 2000 – 2020 (Eurostat, 2019). The reference year of the dataset for the GDP deflator (Index=100) is 2010. Water depths lower than 0.1 m were excluded, considering flood model uncertainties (see Section 2.2). Representative flood depths for buildings and roads were assigned through nearest neighbour interpolation of flood maps applied on the perimeter for buildings (Section 3.3) and on the polyline for roads (Section 3.4).

**3.2 Population**

The number of people affected by coastal flooding was evaluated by considering all the 25 layers of the GHS-POP and ENACT datasets (Section 2.2) through a probabilistic approach. Given that the spatial resolution of the flood model (~ 100 m) is higher than that of the datasets (250 m for GHS-POP, 1 km for ENACT), these were interpolated by using as reference the centre of the cells of the flood map raster (Figure 2). The interpolated values were corrected using the ratio between the cell areas of the

flood map and the datasets. Thus, for each cell of the flood map with non-null values (i.e., the flooded cells), 25 evaluations of the number of people were available. These were used to fit an ECDF for each cell. In order to balance the higher number of layers of the ENACT dataset, weights were assigned: 1/24 for the 24 evaluations based on the ENACT layers, and 1 for the evaluation based on the GHS-layer (i.e., assuming that the value is representative for day/night, for each month). For each cell, the affected number of people was resampled following the probabilistic approach described in Section 3.1, for which a

schematic representation was provided in Figure 2.




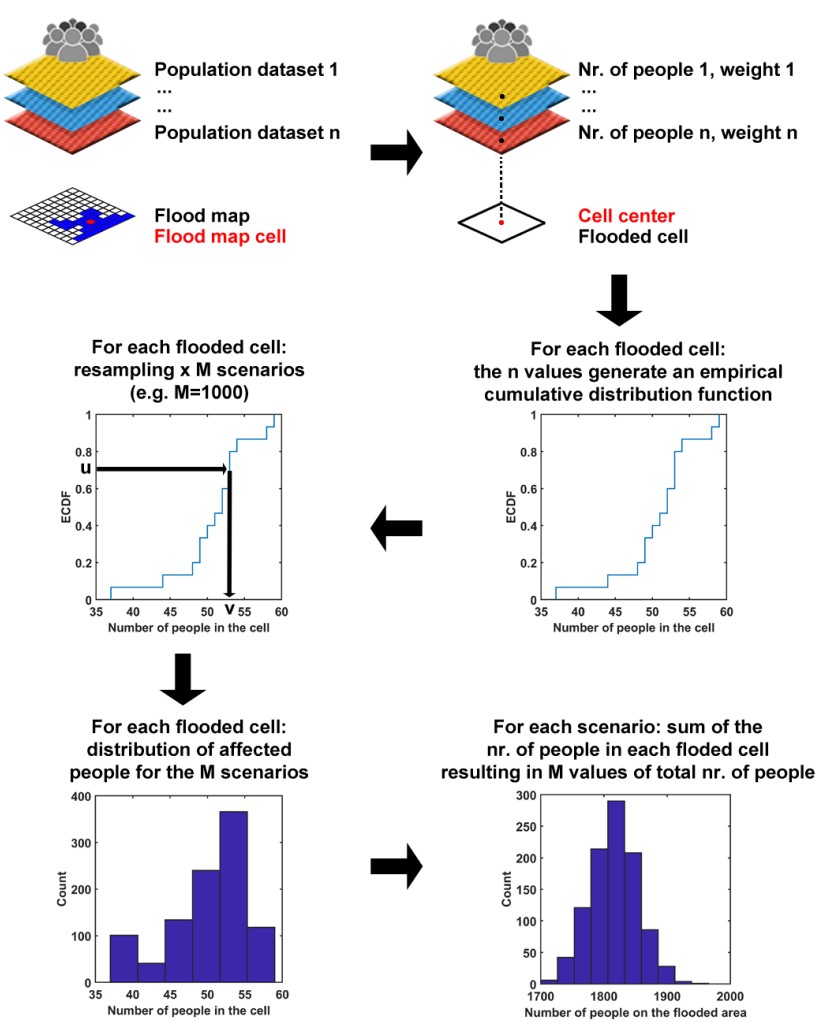

**Figure 2: Scheme of the methodology applied to estimate the number of affected people in the flooded AoIs.**

## 3.3 Buildings

Building damage evaluation relies on an object-based method using flood damage curves (FDCs). Figueiredo et al. (2018) showed that model ensembles offer a useful alternative to deterministic impact assessments, allowing for semi- or fully probabilistic evaluations and considering uncertainty. Duo et al. (2020) also used a similar approach, albeit not fully probabilistic, for assessing damage in Stavanger Harbour, Norway.

Impact models based on FDCs were preferred due to their straightforward implementation and state-of-the-art approach. More complex models (e.g., Dottori et al., 2016; Nofal et al., 2020; Taramelli et al., 2022) were not used due to the lack of required

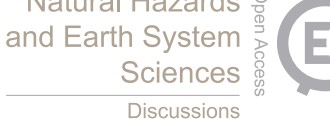

detailed input data at the large scale. Simpler models (e.g., Manselli et al., 2022) do not have the level of details required for this study. The main datasets used were the OSM vector layer of buildings and the CCZ. The OSM layer provides the position and geometric characteristics of the buildings, excluding those with a footprint area less than 20 m$^2$ or identified as Places of Worship. This approach avoids potential outliers in the damage distribution.

The CCZ's level-5 classes were used to categorize OSM buildings, using the dominant class for each element. In a second stage, buildings were reclassified based on macro-classification (residential, commercial, industrial, commercial/industrial, and other) defined in Table A 1 in Appendix A. The CCZ's macro-classification was derived from analyzing the specific classes according to Innerbichler et al. (2021). Assumptions were made to ensure accurate representation of exposed building classes, such as including "Green urban, sports and leisure facilities" (refer to Table B 1 in Appendix B) in the commercial macro-class for leisure and commercial activities in green urban areas. Limitations were discussed in Section 5.3.2. Damage was calculated using FDCs for residential, commercial, industrial, and commercial/industrial macro-classes. No damage was calculated for the "other" macro-class due to the unavailability of damage models. The ensemble approach applies curves described in Table 3 and shown in Figure 3. Seven curves were used for residential buildings; four curves were used for commercial and industrial buildings. For the mixed commercial/industrial category, both commercial and industrial FDCs were applied. The curves were selected based on available technical documentation and scientific publications, ensuring consistency by using damage factors relative to maximum damage. This allowed us to focus on the variability of the vulnerability models (i.e., the FDCs), limiting the uncertainty related to the reference value of the damage factors, that, in other cases, refers to construction or repair costs. The maximum damage from Huizinga et al. (2017) was used for all models. Note that for the mixed class commercial/industrial, defined to consider the aggregated CCZ Level - 5 class 11210 that includes industrial and commercial units, but also public and military units, both commercial and industrial models were applied for a total of eight curves.

Relative damage for each flooded building was calculated using all selected curves for its macro-class, then multiplied by the country-specific object -based maximum damage. The probabilistic evaluation of building damage, total damage in the AoI, and average damage by asset type follows the method described in Section 3.1.

**Table 3: Selected flood damage curves to calculate impacts to the identified buildings: brief descriptions and references are provided along with an indication of the macro-class for which a curve was available (R: residential; C: commerce; I: industry).**

| Flood damage curves | | | Macro-class | | |
|---|---|---|---|---|---|
| Short name | Description | References | R | C | I |
| COMRISK2004 | | Kystdirektoratet (2004) | x | - | x |





| | | | | | |
|---|---|---|---|---|---|
| | Coastal FDCs for the Wadden Sea (estuarine environment) | Vousdoukas et al. (2018a) | | | |
| Hallegatte2011 | Coastal FDCs for Copenhagen | Hallegatte et al. (2011) | x | x | x |
| | | Vousdoukas et al. (2018a) | | | |
| Enghlhardt2019 | Generic FDCs for masonry (IIIb), mixed, concrete and steel (IVb) two-story buildings. | Englhardt et al. (2019) | x | - | - |
| JRC2017 Europe | Generic FDCs for Europe | JRC report and database | x | x | x |
| | | Huizinga et al. (2017) | | | |
| MCM2013 | Coastal FDCs for typical UK properties. Adaptation of the fluvial DDFs with an uplift factor to account for salinity. | Viavattene et al. (2015, 2018) | x | x | - |
| | | Vousdoukas et al. (2018a) | | | |
| Vousdoukas 2018 DDF$_A$ | Coastal FDCs based on small-scale coastal studies | Vousdoukas et al. (2018a) | x | x | x |
| | | Total FDCs for each macro-class | 7 | 4 | 4 |


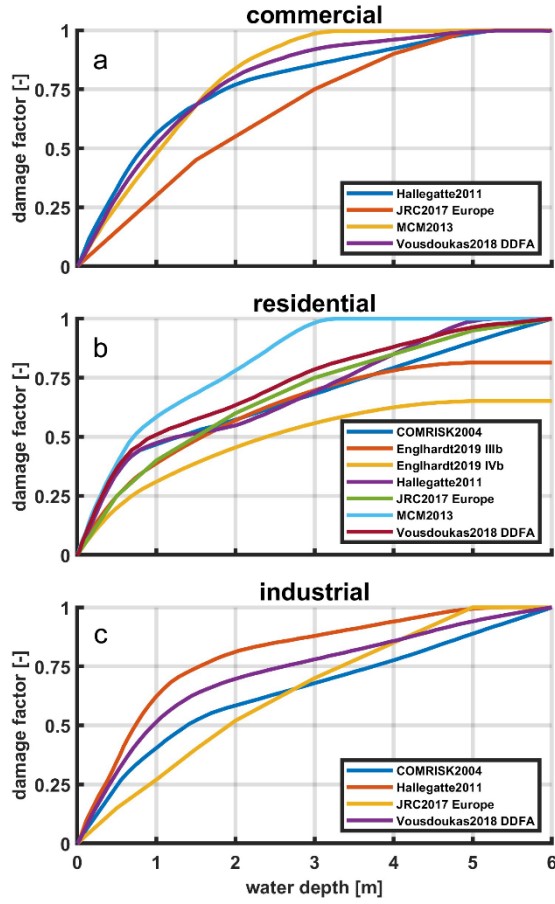

**Figure 3: Overview of the applied flood damage curves for the building types commercial (a), residential (b) and industrial (c).**

**3.4 Roads**

Road impact evaluation uses an object-based method with multiple FDCs, adapted from the work of Van Ginkel et al. (2021) for EU-scale road flood impact assessment. FDCs are based on damage factors relative to construction costs for various road types. The available method was improved for this work by probabilistically resampling literature-based construction cost data from Van Ginkel et al. (2021).

The main dataset used was the OSM roads vector layer, providing position and geometric characteristics of roads. Road macro-classification (see Table B 2 in Annex B) and FDCs (Figure 4) were applied following Van Ginkel et al. (2021). Multiple curves exist for each category, considering road accessory characteristics and hydrodynamic flow conditions. Motorways and Trunks are represented by curves C1 and C2 if highly accessorized (e.g., with street lighting and electronic signalling), C3 and C4 otherwise. Less important roads are represented by curves C5 and C6 (see Table C 1 in Appendix C). All curves were applied, multiplying them by probabilistic resampling of the appropriate construction cost range using ECDFs. An ECDF was

fitted on the literature-based sample of construction costs for motorways from Van Ginkel et al. (2021) and rescaled within defined ranges (Table C 2 in Appendix C) for different road types and accessories. Damage was calculated accordingly,

250 following authors' recommendations (Table C 1 in Appendix C). An overview of applied ECDFs is shown in the Figure 5. The original methodology adjusts costs based on the number of lanes of each road segment, but in this application, default lanes were used (see Table C 2 in Appendix C). The probabilistic resampling considers both multi-FDCs damage factors and construction cost ranges. The probabilistic estimates of total damage and average damage by road type follow the method described in Section 3.1.

255

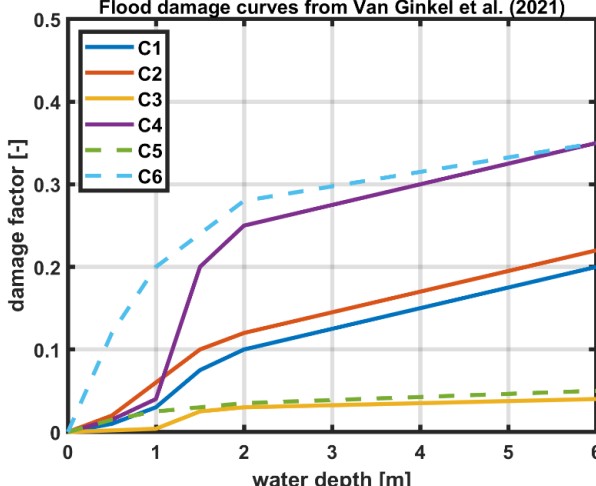

**Figure 4: Overview of the flood damage curves for roads from Van Ginkel et al. (2021).**

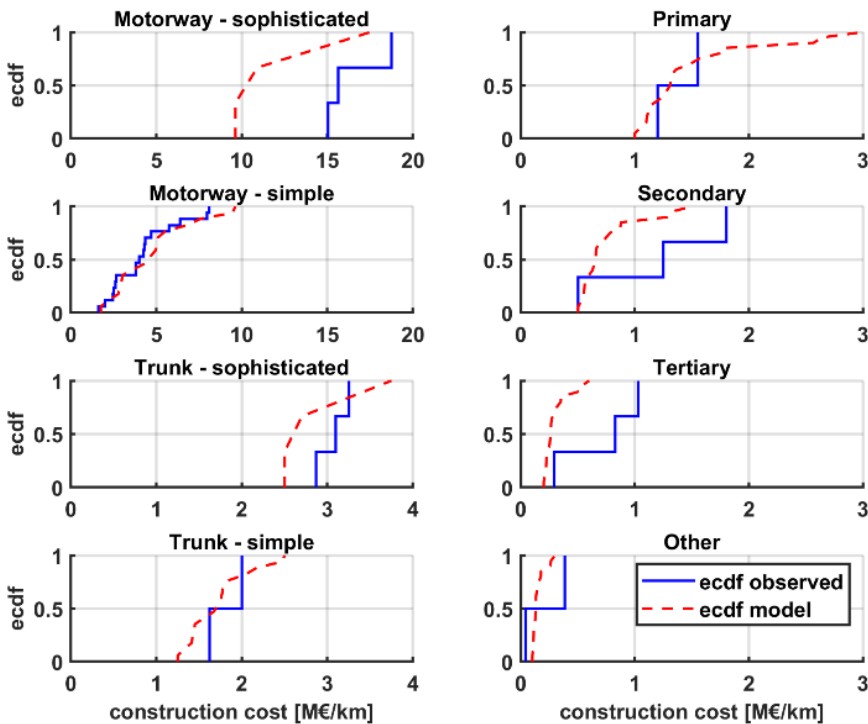

**Figure 5: The literature-based sample of construction costs for motorways from Van Ginkel et al. (2021) was used to calculate an ECDF that was then rescaled based on the ranges of the construction costs defined by the authors for each type of road and level of accessories for the application of the damage model (ecdf model, red dashed line). The curves are compared with the ECDFs of the empirical sample for each type of road (ecdf observed, blue solid line). The recommendations provided by the authors on the correct ranges to apply to each FDCs were followed. The costs are based on 2015 prices for Europe, as reported by Van Ginkel et al. (2021).**

## 3.5 Validation with reported impacts

Validation of the ECFAS Impact approach for the reference cases involved a semi-quantitative, holistic comparison between modelled and reported impact data (see Section 2.4). Quantitative performance evaluations were conducted when reliable quantitative data was reported, while qualitative discussion was given otherwise. In the first case, the comparisons focused on average damage to assets rather than absolute damage due to differences in spatial representation; the probabilistic representation of simulated impacts (95% range; 50% percentile) was considered and reported damages were corrected based on 2020 price levels of the former EU-28, accounting for the event year and country (see Section 2.1). On the other hand, qualitative comparison was applied by evaluating the capacity of the adopted methodology to describe the reported damages. Performance assessment on the AoIs introduced potential bias due to aleatory uncertainties, but these are minimized in large-scale applications (Merz and Thieken, 2009; Wagenaar et al., 2016). For example, in small-scale applications, commercial buildings in an area that is macro-classified as residential (see Section 3.3 for methodology) can lead to overestimation of the flooded buildings' area and average damage per building. To address these issues, comparisons were carefully evaluated for





reliability and representativeness, determining confidence levels. The reliability was defined based on three categories: low, medium, and high, depending on the verifiability of information, whether only general information (low), descriptive (medium) or quantitative and technical (high) information are included. The representativeness was defined based on three categories: low, medium, and high, depending on the scale of the data, whether valid for specific assets, or based on aggregated (from local to large scale) data. Because of the scale of this application, aggregated data at the regional or national level was considered as more representative than punctual or local information. Confidence levels ranged from very low to very high based on the combination of reliability and representativeness (Table 4).

**Table 4: Definition of the confidence based on the reliability and representativeness of the validation.**

|  |  | Reliability | | |
| --- | --- | --- | --- | --- |
|  |  | Only general information [Low] | Only descriptive specific information [Medium] | Quantitative and technical information [High] |
| **Representativeness** | Specific case [Low] | Very low | Low | Medium |
|  | Local scale aggregated data [Medium] | Low | Medium | High |
|  | Large scale aggregated data [High] | Medium | High | Very high |

### 3.6 Comparison with grid-based damage evaluations

To implement a comparison with commonly used impact approaches, and to support the analysis, grid-based impact assessments were implemented for buildings and roads. For consistency with the object-based methods, to implement the grid-based evaluations the chosen reference dataset for LC/LU was the CCZ layer. The flooded cells of the flood maps were considered the basic unit of calculation, to which the most frequent LC/LU class in the cell was assigned.

Based on reclassification of the CCZ class, which is the same applied for the object-based method for buildings (Section 3.3), the damage was calculated for the flooded cell area by applying the FDCs for residential, commercial and industrial buildings, and the LU-based maximum damage provided by Huizinga et al. (2017).



295 The damage to roads was calculated applying the FDCs for infrastructure (roads), and the LU-based maximum damage (25 €/m² in 2010 prices for the entire Europe) provided by Huizinga et al. (2017). The damage was calculated for a fraction of the flooded cell area which was defined for each CCZ class ("Percentage of the road infrastructure" in Table B 1 in Annex B) by adapting the application from Van Ginkel et al. (2021), thus based on guidelines provided by Huizinga et al. (2017) and EEA (2006).

## 4 Results

### 4.1 Reference cases detailed impacts

Detailed results for the Xynthia (France, 2010), Xaver (UK, 2013), and Emma (Spain, 2018) reference cases are presented in this section. Figure 6 displays detailed, disaggregated impact results for buildings, while Figure 7 shows the results for roads, including their uncertainty bands.

305 The residential sector is the most impacted in terms of affected buildings and area in all three reference cases. However, when considering potential total damage to buildings, the residential sector accounts for roughly half of the damage in Xynthia, but around 35% in Xaver and Emma. The remaining damage is primarily associated with commercial or commercial/industrial buildings. Uncertainty in total damage estimates for buildings is generally contained.

Road damage is significantly lower than building damage. Minor roads are most affected, but main roads such as motorways, trunks, and primary roads also experience damage. Notably, no motorways were impacted in the analysed areas. Uncertainty ranges for total road damage, though smaller than building damage ranges, are relevant when compared to the magnitude of the total damage to roads.

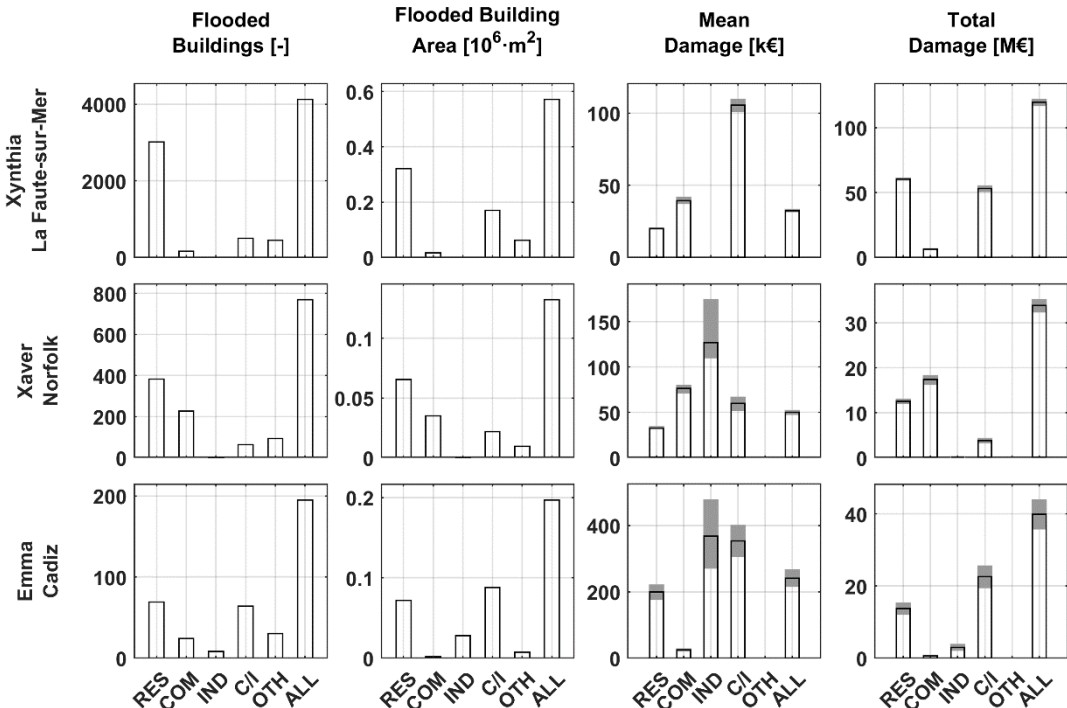

**Figure 6: Overview of impacts to buildings for Xynthia at La Faute-sur-Mer (France, 2010; first row), Xaver at Norfolk (UK, 2013; second row) and, Emma storm at Cadiz (Spain, 2018; third row): number of flooded buildings (deterministic estimate; first column), flooded building area in millions of m² (deterministic estimate; second column), mean damage per asset in thousands of € (probabilistic estimate; third column) and, total damage (in the AoI) in millions of € (probabilistic estimate; fourth column). The results are shown for residential (RES), commercial (COM), industrial (IND), commercial/industrial (C/I), others (OTH) and all (ALL) buildings. Damages are based on average 2020 price levels for EU-28, European Union with 28 Member States; Eurostat, 2019).**




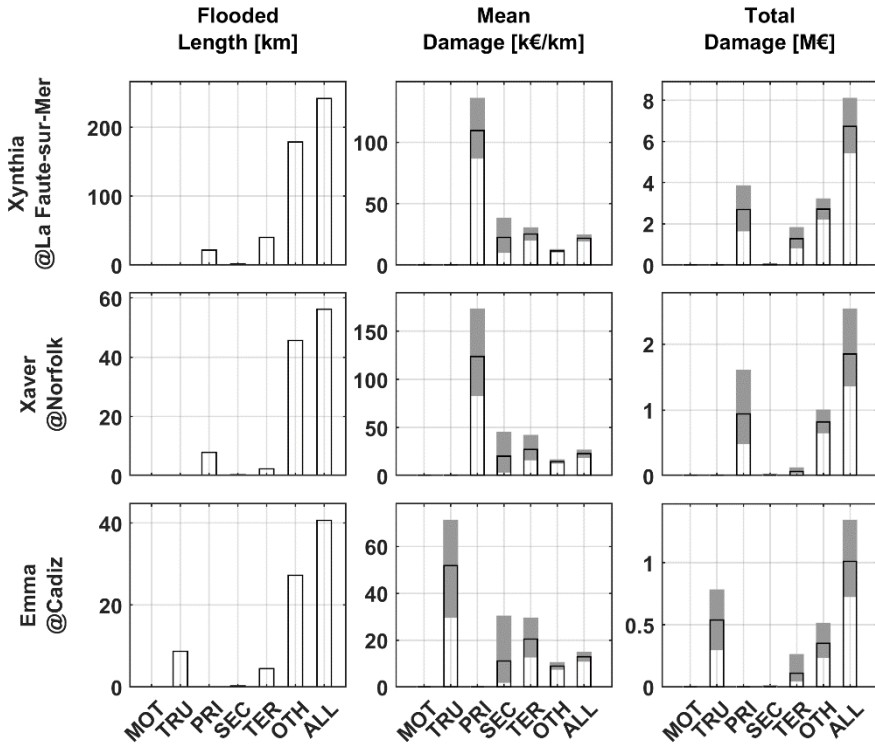

**Figure 7: Overview of impacts to roads for Xynthia at La Faute-sur-Mer (France, 2010; first row), Xaver at Norfolk (UK, 2013; second row) and, Emma storm at Cadiz (Spain, 2018; third row): length of flooded roads in km (deterministic estimate; first column), mean damage per km in thousands of € (probabilistic estimate; second column) and, total damage (in the AoI) in millions of € (probabilistic estimate; third column). The results are shown for motorways (MOT), trunks (TRU), primary (PRI), secondary (SEC), tertiary (TER), others (OTH) and all (ALL) roads. Damages are based on average 2020 price levels for EU-28, European Union with 28 Member States (Eurostat, 2019).**

### 4.2 Overview of impacts for the test cases

The impacts on population, buildings and roads simulated with the ECFAS Impact approach (Sections 3.2, 3.3, and 3.4) are shown in Figure 8 for the analysed test cases. For buildings and roads, the corresponding impact evaluation implemented using a grid-based method (Section 3.6) are also reported in Figure 8, for comparison purposes. The detailed results can be found in Table D 1 in Appendix D.
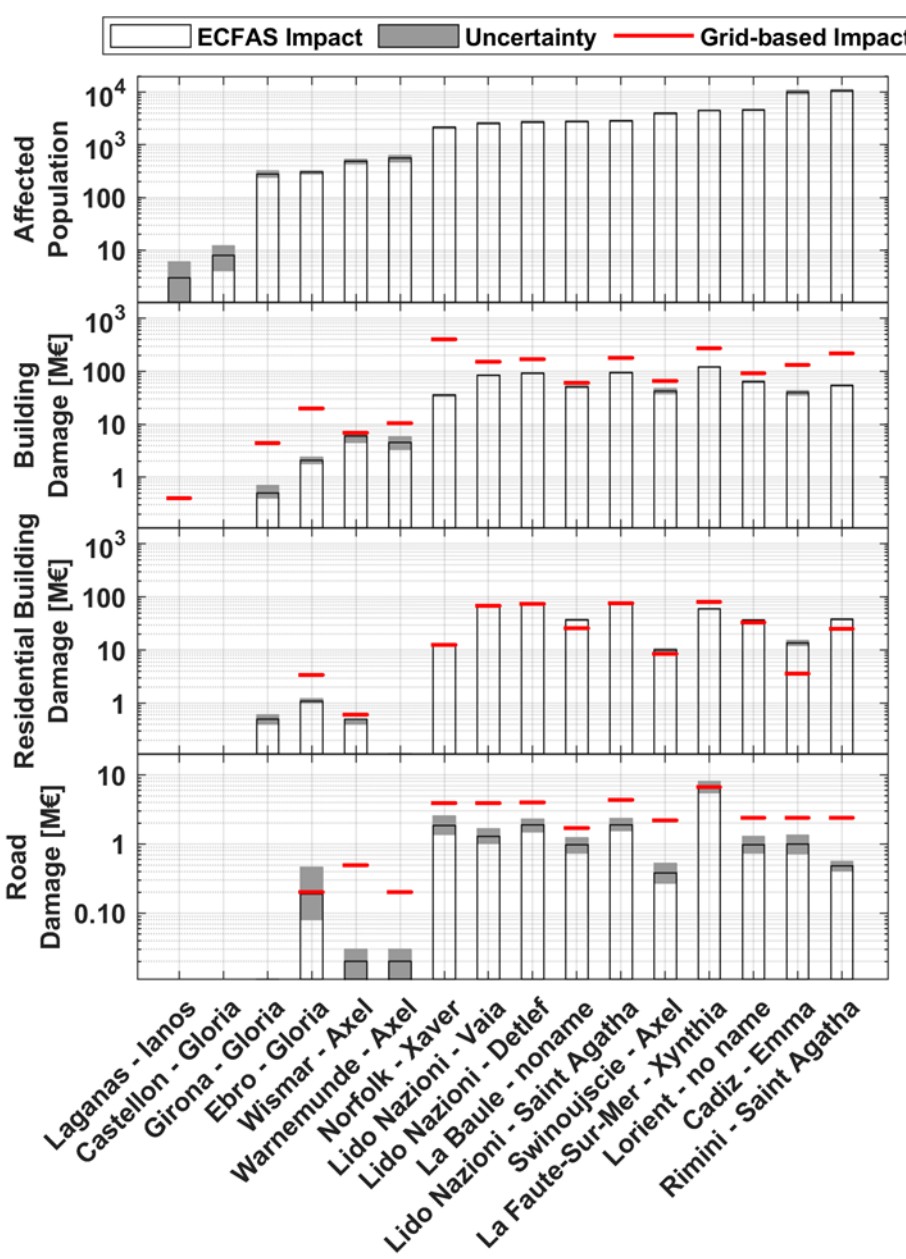

**Figure 8: Impacts on population, buildings and roads simulated with the ECFAS Impact approach for the 16 test cases, and comparison with grid-based methods.**





## 5 Discussion

### 5.1 Validation with reported impacts

Detailed comparisons of simulated damage due to coastal floods with reported data for buildings and roads are summarized in Table 5. Buildings and roads are crucial sectors in terms of flood financial losses, making these comparisons valuable for validation purposes. Reported data vary in type and detail, depending on the country and event significance. Xynthia, for example, raised significant attention and resulted in abundant scientific and governmental information. In the UK, efficient

flood impact collection and analysis, as well as a robust insurance system, provided detailed technical information for events like Xaver. On the other hand, for Emma limited information was available, mainly sourced from media due to a lack of technical reports.

**Table 5: Modelled and reported impact data for the quantitative validation applied for the reference cases. Damages are based on**
**average 2020 price levels for EU-28, European Union with 28 Member States (Eurostat, 2019).**

| Reference case | Sector (type of damage) | Modelled value range 95% (50% percentile) | Reported reference value(s) | Description | Source |
|---|---|---|---|---|---|
| Xynthia (2010) | Residential buildings (average damage per asset) | 19.8 – 20.2 k€ (20.0 k€) | 31.6 k€ (Charente-Maritime) 33.6 k€ (Vendée) 25.4 k€ (All affected areas) | Based on damage to insured properties. Residential damage corrected considering 1.13 households per building (Paprotny et al., 2021) | FFSA-GEMA (2011) |
| | Commercial buildings (average damage per asset) | Commercial: 37.4 - 41.5 k€ (39.5 k€) Com/Ind: 101.2 - 109.6 k€ (105.5 k€) | 60.2 k€ (Charente-Maritime) 31.6 k€ (Vendée) 40.6 k€ (All affected areas) | | |
| Xaver (2013) | Residential buildings (average damage per asset) | 31.7 – 33.9 k€ (32.9 k€) | Fluvial/coastal flood: 6.8-59.3 k€ (lowest-highest) 26.2 k€ (best estimate) Coastal flood only: 35.5 k€ (best estimate) | Based on aggregated data for England and Wales for the winter 2013/14. Data referring to coastal (storm surge) damage only was extrapolated from data | Environment Agency (2016) |
| | Commercial buildings | Commercial: 71.5 - 79.7 k€ | Fluvial/coastal flood: 10.5-125.4 k€ (lowest-highest) | | |





| | | | | | |
|---|---|---|---|---|---|
| | (average damage per asset) | (76.3 k€)<br>Industrial:<br>110.3 - 174.3 k€<br>(126.9 k€)<br>Com/Ind:<br>52.4 - 66.5 k€<br>(60 k€) | 93.5 k€ (best estimate)<br>Coastal flood only:<br>98.7 k€ (best estimate) | from 23 Dec 2013 to 28 Feb. 2014. | |
| | Roads<br>(average damage per km) | All roads:<br>xx – xx k€/km<br>(25 k€/km) | Fluvial/coastal flood:<br>0.67 – 1.62 M€/km (lowest-highest)<br>1.32 M€/km (best estimate)<br>Reported for 155 km of flooded roads | Based on aggregated data for England for the winter 2013/14.<br>Largely uncertain data. | |
| Emma (2018) | Commercial buildings<br>(damage) | Commercial:<br>21.4 – 27.6 k€<br>(24.5 k€)<br>Industrial:<br>272.0 – 477.4 k€<br>(368.8 k€)<br>Com/Ind:<br>306.8 - 400.5 k€<br>(353.5 k€) | 70 k€ for three beach restaurants | Declared by the owner of the commercial activities on the media.<br>Georeferenced information. | Diario de Cadiz (2018a)<br>La Voz del Sur (2018) |
| | Roads<br>(average damage per km) | Tertiary roads:<br>12.9 – 29.3 k€/km<br>(20.4 k€/km) | Carretera Playa de Camposoto (Cadiz, Spain):<br>45.4 k€/km<br>Reported for 1.7 km of flooded road | Retrieved from online news media.<br>Georeferenced information. | Diario de Cadiz (2018b) |

### 5.1.1 Xynthia storm, La Faute-sur-Mer (France), 2010

Buildings. The modelled damage for residential buildings underestimates reported values by a factor of 0.6 when compared to disaggregated data. When compared to aggregated (all areas) reported damage, the simulated damage underestimates average damage by a factor of 0.8. Note that the simulated damage includes both structure and content. Content damage is estimated to be roughly 30% of total damage (André et al., 2013; Paprotny et al., 2021). The simulation does not consider building



collapse, which was an important aspect for this test case: extensive damage led to destruction of properties and compensation by the government (~1500 houses at an average of €150,000 per house) (Kolen et al., 2013).

The average simulated damage for commercial and mixed (commercial/industrial) categories differs significantly due to
varying footprint areas. For the mixed category, 27% of the flooded building area derives from assets larger than 1000 $m^2$, while commercial buildings are all smaller than 1000 $m^2$. Reported damage for professional properties aligns with simulated damage for commercial buildings by a factor of 0.65-1.25. However, for the mixed category, the comparison shows factors higher than 1.75.

*Roads*. Quantitative information on road impacts was limited. Government reports mention significant damages, while media
and other sources show erosion, debris deposition, and asphalt damage. The Route de la Tranche-sur-Mer (Figure 9a) experienced significant erosion outside the AoI. A quick assessment considered it as a tertiary road with a simulated flood of approximately 1 m. Damage for the 375 m segment ranged from 2.1 - 43.4 k€ (5.5 - 115.8 k€/km). Calculated damage factors for the AoI were 0.6 - 28.3%, with only 20% of roads showing relative damage above 15%. Higher values above 11% (van Ginkel et al., 2021) matched reported damages (Figure 9a), suggesting that the higher portion of the distribution (75-97.5%)
better represents road damage. The model considers both low and high flow conditions, while reported damages mainly relate to high flow.

### 5.1.2 Xaver storm, Norfolk (United Kingdom), 2013

*Buildings*. Simulated damage in the residential sector aligns with the reported range for fluvial and coastal floods (Environment Agency, 2016). However, it slightly exceeds the reported best estimate. When considering disaggregated data for coastal floods
only, the average damage matches the simulation. The reported estimation, extrapolated from data from 23 December 2013 to 28 February 2014, likely underestimates coastal flood damage for the Xaver event. Overall, the impact approach for residential buildings appropriately represents the magnitude of the average damage for coastal flooding. However, a tendency to overestimate the average damage must be underlined, as seen in the comparison with the reported best estimate for fluvial and coastal floods. The probabilistic-based approach accounts for this, including uncertainties in vulnerability models.
Nevertheless, an overestimation of residential damage, even in magnitudes, is expected (Molinari et al., 2020). In this specific case, the approximation factor is 0.9-1.25 when compared to the reported best estimate for fluvial and coastal floods.

The average simulated damage significantly differs between the commercial and mixed categories, but it is comparable in magnitude. Only one industrial building in the flooded area incurs damage (~127 k€). The model underestimates damage for commercial buildings by a factor of 0.7-0.8. The evaluation for the mixed category also falls below the reported best estimate.
When considering disaggregated data for coastal floods only for business properties, the underestimation is slightly emphasized. However, all simulated damages align with the reported range for riverine and coastal floods.

*Roads*. The comparison between simulated and reported average damage reveals a significant discrepancy, with simulated damage being two orders of magnitude lower than reported values (for fluvial and coastal floods). The reported information on road impacts carries large uncertainty, as acknowledged by the authors of the report (Environment Agency, 2016). The lack



of detailed context information in the report raises doubts about the accuracy of the reported length of flooded/affected roads. Moreover, the reported examples primarily focus on damages to motorway and trunk roads, suggesting that the reported average damage may be more representative of those road types. In contrast, the simulated results primarily represent primary and other roads (see Figure 7). Simulated average damage for primary roads is approximately one order of magnitude higher than other road classes. Assuming all flooded roads as trunks, the average simulated damage ranges from 129.2 to 141.8 (135.5)

k€/km. Similarly, for motorways, it ranges from 468.0 to 516.3 (491.9) k€/km. This assessment supports the notion that the reported data may better reflect the average damage for motorways and trunks.

By analysing reported information (see Section 3.5), the type of damages affecting roads can be determined and qualitatively compared with the estimated damage from the applied model. Figure 9 (b and c) provides examples for this test case, where debris deposition represents the main physical impact on minor roads (primary, secondary, tertiary, other). Cleaning operations

account for most of the financial damage, while repair works typically pertain to the regular maintenance due to lower maintenance standards compared to motorways and trunks (van Ginkel et al., 2021). These reported damages indicate that the flooding had low flow velocities, whereas the model considers damage curves for both low and high flow velocities. Therefore, it is reasonable to assume that the simulated damage for roads overestimates the overall damage, and the lower half of the sample (percentiles: 2.5%-50%; relative damage < 5%) better represents the actual road damage. The construction cost used

to calculate absolute damage introduces some uncertainty, but this is addressed through probabilistic application.

### 5.1.3 Emma storm, Cadiz (Spain), 2018

*Buildings*. The high simulated damage for the residential sector could be related to the existence of large residential buildings. None of the sources of information analysed refer to damage to residential properties for this reference case, which could lead to the conclusion that residential buildings were not affected by the flooding, although the analysed resources do not represent

official reports.

Considering the commercial sector, the comparison was implemented by analysing a single case of a beach restaurant. This building was repeatedly flooded during the event (Figure 9e), as confirmed by news, videos and the qualitative analysis of the data from a video monitoring system in the area (Montes et al., 2018). The simulated damage for the beach restaurant is estimated between 52.5 and 99.5 k€ (95% probability), and the 50% percentile is 78.8 k€. The estimated damage reported by

the owner of one of the beach restaurants in the area considered three beach restaurants. Nonetheless, the other two properties did not suffer significant damages, and it is reasonable to assume that most of the reported damage refers to the former. By taking this aspect into account, and the fact that the owner may have overestimated the damages, the comparison between the average simulated and the reported damage shows no significant differences. The approximation factors vary in the range of 0.75-1.4.

*Roads*. The damage reported for the Carretera Playa de Camposoto is higher than the upper limit of the simulated range for tertiary roads. However, it is comparable with the simulated damage for the specific road: 5.8 - 121.3 (38.9) k€/km. The corresponding simulated relative damage is 2.6-21.3%. The results are in line with the observed damage (i.e., mainly cleaning




costs and possible minor damage to asphalt; Figure 9d). The lower limit of the simulated damage is expected to represent those

cases where only cleaning cost is needed.


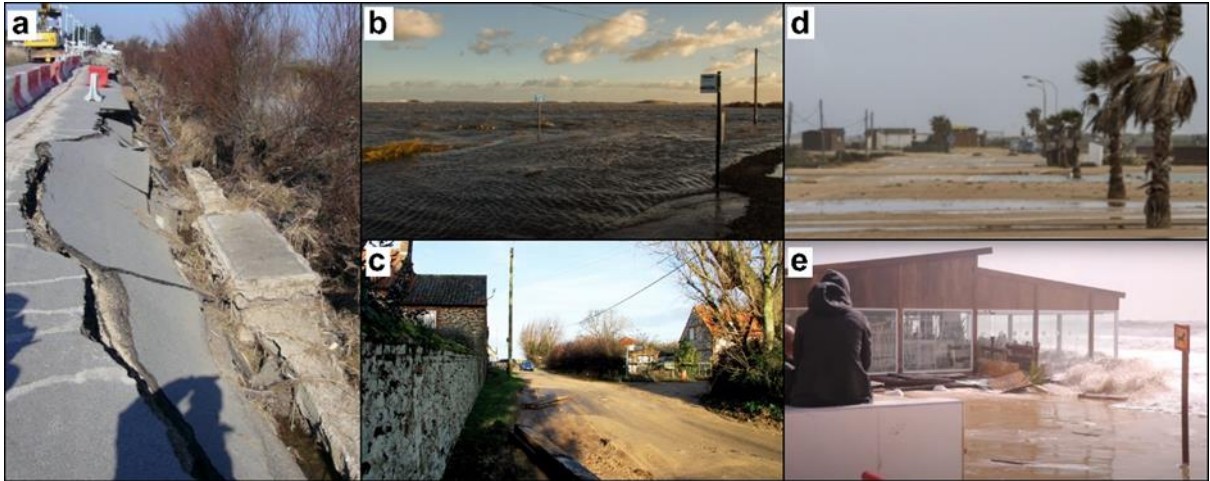

**Figure 9: Examples of damage to roads: (a) Route de la Tranche-sur-Mer in the area of La Faute-sur-Mer (France) after the Xynthia event in February 2010; (b) Coast Rd (Salthouse, Holt) and (c) Beach Rd (Holme-next-the-Sea, Hunstanton) in the Norfolk (UK) area after the Xaver event in December 2013; (d) Carretera Playa de Camposoto in the south of the city of Cadiz (Spain) after the**
**Emma event in March 2018. The images were retrieved from the sources of information collected in the ECFAS database of extreme events and test cases (Souto Ceccon et al., 2021).**

### 5.1.4 Confidence

The comparisons for the reference cases were assessed based on confidence levels (Section 3.5). For residential buildings, the

Xynthia and Xaver storms showed underestimations (max. factor 2) and overestimations (max. factor 1.3), with good

agreement in magnitude. No validation was possible for residential buildings in the Emma reference case. Comparisons for

commercial buildings showed underestimates (max. factor 2) and overpredictions (max. factor 3), with appropriate magnitude

estimates. Road comparisons generally agreed with reported damages, with slight overestimations expected.

High confidence was assigned to the Xynthia reference cases validation, based on aggregated data from national insurance and

scientific reports. The Xaver reference case had medium-high confidence due to reliable national technical reports aggregated

that were considered more representative for evaluating the performance of an impact model to be applied at the large scale.

The Emma reference case had low confidence due to limited data availability and representativeness. Commercial building

validation relied on specific news information, while road comparison was limited to a 1.7 km segment in Cadiz.

### 5.2 Comparison with grid-based damage evaluations

The comparison in Figure 8 showed that object-based evaluations of the ECFAS Impact approach generate lower results than

grid-based methods for total damage to buildings and roads. The latter often report values that are two or more times, or even





one order of magnitude, higher. Interestingly, the differences never exceed one order of magnitude, and the grid-based damage to residential buildings showed a general agreement with the corresponding object-based evaluations, with some exceptions. For damage to buildings, in fact, the residential FDC from Huizinga et al. (2017) applied for the grid-based estimates

approximates the average behaviour of the set of curves applied by the ECFAS Impact approach (Figure 3) for residential buildings. Moreover, the maximum damage for both methods is retrieved from the same source (Huizinga et al., 2017). Although overestimation can be expected for the grid-based assessment, the overestimates of the building flooded area is partially balanced by the lower (LU-based) maximum damage applied. In this context, the ECFAS Impact approach represents a more reliable method for refined damage estimates. The object-based approaches outperform the grid-based ones in terms of

resolution and detail of the assessment, although no conclusion can be drawn on the performance of grid-based methods when compared with reported data. Indeed, implementing the comparisons as described in Section 3.5 would not be feasible with grid-based results because of the nature of the methodology (e.g., it is not possible to estimate the number of affected assets without the availability of specific information).

## 5.3 Limitations

### 5.3.1 Population

The evaluation of affected people provides an estimate of individuals directly exposed to the flood. Uncertainties arise from temporal differences between datasets (ENACT: 2011; GHS: 2015) and the flood event's reference year, as well as spatial resolution discrepancies between the flood map and datasets. Uncertainty due to the input datasets are expected for these type of assessment (Lichter et al., 2011). The probabilistic resampling partially accounts for it.

It also accounts for seasonal and day/night variability. Timing-based assessments of the affected population would be more appropriate for operational applications. Certainly, it would represent a refinement of the assessment from a deterministic point of view. Comparisons between GHS-POP and ENACT datasets for the number of people in the affected areas show minor variations (Figure 10; horizontal axes in logarithmic scale). Exceptions exist, like Castellon (Spain), where ENACT's low resolution makes it statistically unreliable. Overall, similar numbers of affected people are identified, with acceptable variations

within the same magnitude. Probabilistic implementation was preferred due to these reasons.
Constructing ECDFs at the flood map cell scale involved applying different weights to datasets. Equal weights would favour ENACT, so using different weights aims to homogenize representativeness. This weighting method has a significant impact on the evaluation. Alternative solutions may require assumptions on dataset uncertainty, generating values to feed the ECDF, but these introduce additional uncertainties.

Validating the reliability of simulated affected population numbers using reported figures is challenging, as reported data mostly focus on casualties, injuries, and hospitalizations. These factors depend not only on human presence but also on early warning systems and emergency response efficiency. Additional considerations could involve evacuated households, long-term flood-related illnesses, or other indirect impacts to estimate the number of affected people.





Vulnerability-based evaluations can enhance flood risk assessment, but large-scale implementation is hindered by the need for
detailed socio-economic, cultural, and governance data (Thomas et al., 2019).

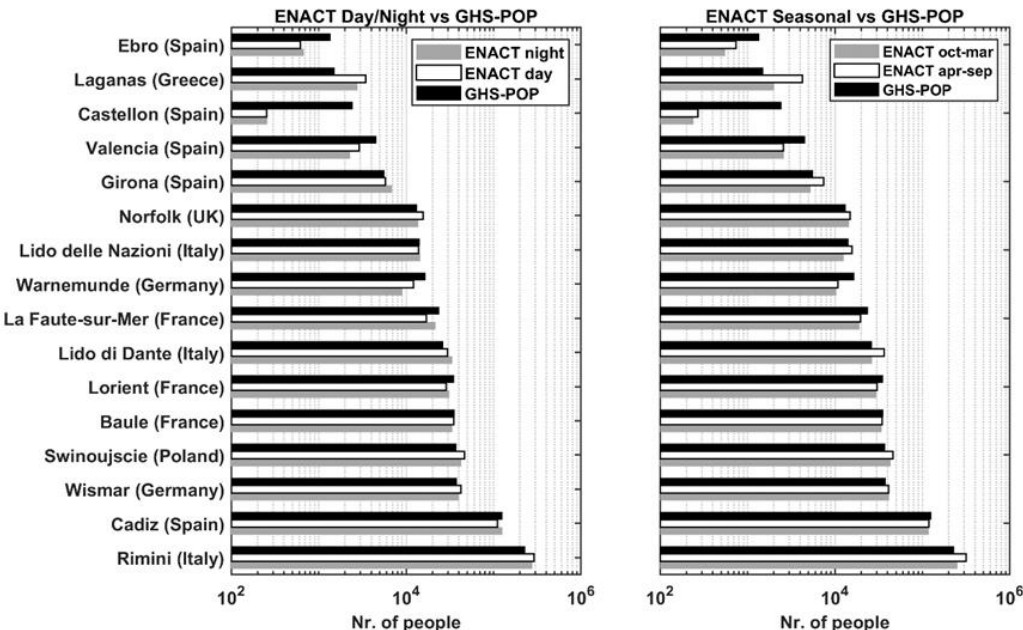

**Figure 10: Overview of the population on the AoIs of the sites' test cases: the GHS-POP estimate is compared with the ENACT
estimate for the day-time and night-time (left box) and with the ENACT estimate for summer (April-September) and winter**
**(October-March) seasons (right box). The number of people is represented using a logarithmic scale.**

**5.3.2 Buildings**

OSM provides reliable building coverage for large-scale evaluations at the EU level. However, quality control at specific sites
revealed some coverage gaps (e.g., Castellon site in Spain), leading to underestimations of building damage. Nevertheless, this
limitation is considered non-critical as OSM is regularly improved and updated.

The macro-classification is based on CCZ layer representativeness analysis. OSM buildings were found on beaches classified
as open spaces in the CCZ layer (sandy, 62111; shingle, 62112), representing beach facilities and economic activities, such as
tourism (e.g., Emilia-Romagna coast Italian sites: Lido delle Nazioni, Lido di Dante and Rimini). Hence, beach related CCZ
classes were included in the commercial macro-class, providing a practical solution for commercial buildings located on
beaches without specific commercial classification in the CCZ layer.

Numerical interpolation of data introduces limitations. The interpolation method for representative flood depth calculation can
influence the number of flooded buildings and the building macro-classification. These aspects refer to the aleatory component
of the uncertainty and should be therefore contained when modelling large-scale events (see Section 3.5).

The application utilizes damage factors and object-based maximum damage (provided by Huizinga et al., 2017), accounting
for country-based GDP. However, this simplified approach may not capture intra-country variability, and maximum damage-



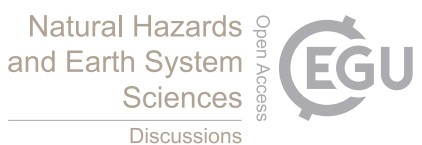

based curves generally overestimate flood damage to buildings (Molinari et al., 2020). To improve the assessment, robust national, regional, or municipality-based damage curves can be generated, as recently demonstrated by Martínez-Gomariz et al. (2020) for Spain.

The multi-model ensemble implementation based on Figueiredo et al. (2018) has limitations due to the number of models and other factors discussed in previous studies (e.g., Duo et al., 2020). In this case, the number of models applied for each building macro-class is limited due to the scale of application of the impact assessment. Despite limitations, multi-model ensembles demonstrate better predictive skills compared to single-model (deterministic) assessments (Figueiredo et al., 2018).

### 5.3.3 Roads

Limitations exist regarding the numerical interpolation of data, as discussed previously for building impacts.

The macro-classification used is adapted from Van Ginkel et al. (2021). Negligible uncertainty is expected from this reclassification due to the coverage of OSM roads dataset, particularly for important roads. However, there may be some uncertainty for less significant roads.

In this case, probabilistic resampling assumes that the empirical construction cost distribution for motorways is applicable to other road types. The comparisons shown in Figure 5 supports this hypothesis. The main difference is observed for highly accessorized motorways due to a lack of observations in the literature-based sample within a specific cost range (8.2 and 14.9 M€/km; 2015 prices).

Probabilistic resampling is applied at two levels: for input data on construction costs and for multiple FDCs applied to each road class. This provides an evaluation of uncertainty in both construction costs and FDCs.

The approach uses the default number of lanes for each road segment. While this may introduce uncertainty, it should be limited, particularly at a large scale (aleatory uncertainty, see Section 3.5), and it is accounted for by probabilistic resampling of construction costs.

It is important to note that certain aspects, such as infrastructure failure or damage from compound hazards (e.g., pluvial, landslides), are not addressed by the method of Van Ginkel et al. (2021), as recently demonstrated by Koks et al. (2022).

### 6 Conclusions

The ECFAS Impact approach assesses flood direct impacts on population, buildings, and roads in European coastal areas. Developed within the EU H2020 ECFAS Project, the methodology incorporates object-based and probabilistic evaluations. The presented approach was validated against reported impacts for the Xynthia storm at La Faute-sur-Mer (France) in 2010, the Xaver event at Norfolk (UK) in 2013, and the Emma storm at Cadiz (Spain) in 2018; and it was tested against grid-based approaches on 16 test cases across Europe.

Key findings indicate that the approach provides:

- valuable estimates for affected populations to address prevention and preparedness activities;





- reliable damage assessments for buildings and roads that can feed further evaluations of risk scenarios, including cost-benefit analysis of DRR strategies;

- improved accuracy for damage to buildings and roads compared to traditional grid-based approaches.

The ECFAS Impact approach improves upon the EU-scale operational approach for riverine flood warnings (EFAS) by utilizing detailed recent datasets and probabilistic methods. Additionally, the adoption of object-based methods for buildings and roads improves the detail and reliability of the simulated impacts, moving from the meso-scale to the micro-scale analysis. The approach presented in this work is part of the ECFAS system, which can extend the capabilities of the Copernicus Emergency Management System for coastal flood early warnings, complementing EFAS for riverine floods. Future developments aim to enhance population assessment incorporating vulnerability and risk to life estimates and refine damage 540 and uncertainty evaluation for buildings and roads.



## Appendix A. Datasets

**Table A 1: Overview of the dataset used for the impact assessment. The data is available through the Zenodo platform (Ieronymidi and Grigoriadis, 2022; https://zenodo.org/records/7319270).**

| Dataset | Type | Ref. year | Res. | Nr. of layers | Description | Link |
|---|---|---|---|---|---|---|
| Global Human Settlement Population Grid (GHS-POP R2019A) | Raster | 2015 | 250 m | 1 | Distribution of population (nr. of people per cell) for the year 2015 | https://ghsl.jrc.ec.europa.eu/ghs_pop2019.php |
| ENACT 2011 Population Grid (ENACT-POP R2020A) | Raster | 2011 | 1 km | 24 | Distribution of population (nr. of people per cell) during night- and day-time, for each month of the year 2011 | https://data.jrc.ec.europa.eu/dataset/be02937c-5a08-4732-a24a-03e0a48bdcda |
| Open Street Map (OSM) | Vector | 2021 | Various | 4 | Buildings (polygons) Roads (polylines) Railways (polylines) POIs (points and polygons) | www.openstreetmap.org |
| Copernicus Coastal Zone (CCZ) | Vector (polygons) | 2018 | Minimum mapping 0.5 ha 10 m | 1 | Land use classification for the EU coastal area | https://land.copernicus.eu/local/coastal-zones |



## Appendix B. Macro-classifications defined for buildings and roads

**Table B 1: Macro-classification of the type of building based on the Copernicus Coastal Zone layer classes and assumed percentage of road infrastructure for each relevant class.**

| ECFAS macro-category | Copernicus Coastal Zone layer | | Percentage of road infrastructure |
|---|---|---|---|
| | 5-digit code | Class | |
| Residential | 11110 | Continuous urban fabric (IMD* ≥80%) | 18 |
| | 11120 | Dense urban fabric (IMD ≥30-80%) | 12 |
| | 11130 | Low density fabric (IMD <30%) | 6 |
| Commercial | 12350 | Marinas | 40 |
| | 14000 | Green urban, sports and leisure facilities | 10 |
| Industrial | 11220 | Nuclear energy plants and associated land | 21 |
| | 12310 | Cargo port | 40 |
| | 12330 | Fishing port | 40 |
| | 12370 | Shipyards | 40 |
| | 13110 | Mineral extraction sites | 21 |
| | 13120 | Dump sites | 21 |
| | 13130 | Construction sites | 21 |
| Commercial/Industrial | 11210 | Industrial, commercial, public and military units (other) | 21 |
| | 12340 | Naval port | 40 |
| Other | 12100 | Road networks and associated land | 100 |
| | 12320 | Passenger port | 40 |
| | 12360 | Local multi-functional harbours | 40 |
| | 12400 | Airports and associated land | 40 |
| | - | All other classes | - |

**Table B 2: Macro-classification of the type of roads based on OSM classes.**

| Road type | OSM class |
|---|---|
| | |



| Motorway | motorway, motorway_link, motorway_junction |
| --- | --- |
| Trunk | trunk, trunk_link |
| Primary | primary, primary_link |
| Secondary | secondary, secondary_link |
| Tertiary | tertiary, tertiary_link |
| Other | unclassified, residential, living_street, service, pedestrian, bus_guideway, escape, raceway, road, cycleway, construction, bus_stop, crossing, mini_roundabout, passing_place, rest_area, turning_circle, traffic_island, yes, emergency_bay |
| Track | track, unsurfaced, corridor, trail, footway, path |
| None | none, bridleway, steps, proposed, elevator, emergency_access_point, give_way, speed_camera, street_lamp, services, stop, traffic_signals, turning_circle, toll_gantry, stop, disused, dummy, planned, razed, abandoned |





## Appendix C. Additional information on impact to roads

**Table C 1: Flood damage curves from Van Ginkel et al. (2021): characteristics of roads and hydrodynamic flow conditions.**

| Curve ID | Road type | Road accessories | Hydrodynamic flow conditions |
|---|---|---|---|
| C1 | Motorway Trunk | Sophisticated accessories (i.e., curves to be applied with the upper half of the provided range of construction costs; Table C 2) | Low |
| C2 | | | High |
| C3 | | Simple roads (i.e., curves to be applied with the lower half of the provided range of construction costs; Table C 2) | Low |
| C4 | | | High |
| C5 | Primary Secondary Tertiary Other | No embankments | Low |
| C6 | | | High |

**Table C 2: Default number of lanes and construction cost ranges by road type from Van Ginkel et al. (2021).**

| Road type | Default nr. of lanes per road segment [directions x lanes] | Min. constr. cost [M€/km] | Max. constr. cost [M€/km] |
|---|---|---|---|
| Motorway* | 1x2 | 1.75 | 17.5 |
| Trunk* | 1x2 | 1.25 | 3.75 |
| Primary | 2x1 | 1.0 | 3.0 |
| Secondary | 2x1 | 0.5 | 1.5 |



| Tertiary | 2x1 | 0.2 | 0.6 |
|---|---|---|---|
| Other | 1x1 | 0.1 | 0.3 |






**Appendix D. Overview of the results for each test case**

**Table D 1: Overview of impacts on population, buildings and roads for the test cases.**

| Test Case | Nr. Of people inside flooded area | Flooded Building Area [m²] | Nr. Of flooded buildings | Buildings Total Damage [M€] | Nr. Of affected residential buildings | Residential Damage [M€] | Roads Length [km] | Roads Total damage [M€] |
|---|---|---|---|---|---|---|---|---|
| Baule – No name | 2924 – 3131 (3028) | 238957 | 1436 | 48.7-52.2(50.5) | 1296 | 36.1 - 37.9 (37.0) | 36.42 | 0.74 - 1.24 (0.97) |
| La Faute-sur-Mer – Xynthia | 4432 - 4612 (4521) | 571821 | 4130 | 117.2-121.9(119.6) | 3016 | 59.6 - 60.9 (60.3) | 242.03 | 5.47 - 8.09 (6.74) |
| Lorient – No name | 4421 - 4666 (4544) | 357189 | 2453 | 60.6-66.5(63.6) | 2058 | 36.3 - 37.1 (36.7) | 52.48 | 0.74 - 1.28 (0.97) |
| Warnemunde – Axel | 478 - 636 (558) | 33769 | 35 | 3.0-5.6(4.3) | 6 | 0.1 - 0.1 (0.1) | 2.75 | 0.01 - 0.03 (0.02) |
| Wismar – Axel | 433 - 537 (486) | 56449 | 76 | 4.5-7.1(5.9) | 33 | 0.4 - 0.5 (0.5) | 2.65 | 0.01 - 0.03 (0.02) |
| Laganas – Ianos | 1 - 6 (3) | 133 | 1 | 0.0-0.0(0.0) | - | - | 0.03 | - |
| Lido delle Nazioni – Saint Agatha | 2755 - 2967 (2862) | 295138 | 1423 | 93.2-97.1(95.2) | 1089 | 75.3 - 77.3 (76.3) | 72.28 | 1.56 - 2.34 (1.91) |


| | | | | | | | | |
|---|---|---|---|---|---|---|---|---|
| Lido delle Nazioni – Vaia | 2470 - 2675 (2573) | 227942 | 1162 | 82.2-85.8(84.1) | 954 | 66.2 - 67.9 (67.1) | 50.97 | 1.02 - 1.66 (1.30) |
| Lido delle Nazioni – Detlef | 2614 - 2816 (2713) | 265682 | 1315 | 90.4-94.0(92.2) | 1014 | 73.0 - 75.0 (74.0) | 68.24 | 1.51 - 2.32 (1.89) |
| Rimini – Saint Agatha | 10100 - 11159 (10635) | 373525 | 1200 | 51.9-55.0(53.5) | 786 | 36.7 - 38.7 (37.7) | 52.59 | 0.41 - 0.56 (0.48) |
| Swinoujscie – Axel | 3746 - 4142 (3953) | 140311 | 268 | 35.7-44.3(40.2) | 103 | 9.4 - 10.7 (10.0) | 24.28 | 0.27 - 0.53 (0.38) |
| Castellon – Gloria | 4 - 12 (8) | - | - | - | - | - | - | - |
| Ebro – Gloria | 279 - 324 (301) | 12235 | 91 | 1.7-2.2(2.0) | 54 | 1.0 - 1.2 (1.1) | 9.59 | 0.08 - 0.46 (0.19) |
| Girona – Gloria | 243 - 317 (277) | 5702 | 8 | 0.4-0.7(0.5) | 6 | 0.4 - 0.6 (0.5) | 1.32 | 0.00 - 0.01 (0.01) |
| Norfolk – Xaver | 2041 - 2244 (2139) | 132491 | 770 | 32.5-35.2(33.9) | 383 | 12.1 - 13.0 (12.6) | 56.18 | 1.37 - 2.53 (1.85) |
| Cadiz - Emma | 9261 - 10952 (10103) | 197006 | 195 | 35.9-43.9(39.9) | 69 | 12.2 - 15.2 (13.8) | 40.63 | 0.73 - 1.34 (1.01) |



**Code and data availability.** The data and code related to this work and produced during the EU H2020 ECFAS project (GA 101004211; www.ecfas.eu) can be accessed through the Zenodo platform: Impact Tool (Duo et al., 2022; https://doi.org/10.5281/zenodo.7489035); Pan-EU Flood Catalogue (Le Gal et al., 2022; https://doi.org/10.5281/zenodo.7488978); Pan-EU Impact Catalogue (Duo et al., 2022; https://doi.org/10.5281/zenodo.6951527). The data presented in this study are available on request from the corresponding
author.

**Author contribution.** Enrico Duo: Conceptualisation, Methodology, Software, Validation, Formal analysis, Investigation, Writing - Original Draft, Visualization. Juan Montes Perez: Methodology, Software, Validation, Formal analysis, Investigation, Writing - Original Draft. Marine Le Gal: Resources, Writing - Review & Editing. Tomas Ferńandez Montblanc:
Resources, Writing - Review & Editing. Paolo Ciavola: Writing - Review & Editing, Supervision, Funding acquisition. Clara Armaroli: Writing - Review & Editing, Supervision, Funding acquisition.

**Competing interest.** The authors declare that they have no known competing financial interests or personal relationships that could have appeared to influence the work reported in this paper.


**Acknowledgements.** The authors are thankful to Paulo Cabrita, Paola Emilia Souto Ceccon, Maialen Irazoqui, Vera Gastal, Sebastien Delbour, Dionysis Grigoriadis, and Emmanouela Ieronymidi, for their support to the work described in this paper. This work received funding from the H2020 European Project ECFAS (A proof-of-concept for the implementation of a European Copernicus coastal flood awareness system, GA n° 101004211; www.ecfas.eu). Juan Montes has a postdoctoral
contract Margarita Salas at the University of Cadiz from the Ministry of Universities of Spain, funded by the European Union-NextGenerationEU. Marine Le Gal benefited from the "Go for IT" grant (Area 04 - Scienze della Terra) from the Fondazione CRUI under the responsibility of Prof. Paolo Ciavola.

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
