# Peer review of "Validated probabilistic approach to estimate flood direct impacts on the population and assets on European coastlines"

_Natural Hazards and Earth System Sciences, 2023_

## Author Response (AR1)

Juan Montes
Earth Sciences Department, INMAR, University of Cadiz
23/05/2024

Dear Editor Robert Sakic Trogrlic,

We thank you for your time and for the opportunity to submit the reviewed version of this manuscript entitled "Validated probabilistic approach to estimate flood direct impacts on the population and assets on European coastlines".

The present document contains detailed answers to the comments made by the two reviewers and the author's track-changes file to show the modifications made following the review. In addition, a reference to the part of the text that has been modified as a result of the comments is included in the replies. The lines' numbers refer to the new track-changes manuscript version.

We hope that the answers will clarify the concerns raised during the review and improve the document to be considered for acceptance.

Many thanks,
Juan Montes

**Response to the reviewer 1**

Dear referee, many thanks for your time and feedback. This reply contains the detailed answers to the comments provided in the first reply, modified where necessary according to the changes made in the text, which is also attached to show the changes. A reference to the part of the text that has been modified as a result of the comments is also included for each comment.

**1.      Line 33: Indirect impacts also have short-term consequences.**

As the way it is written may lead to confusion, the paragraph was corrected in L33-36 to clarify that indirect impacts also have short-term consequences.

**2.      Line 35/36: Sentence a bit vague, not clear why you use the word "instead".**

It was corrected to improve the connection between sentences and make the paragraph easier to understand, as the sentence was a bit vague. Modifications are made on L33-36.

**3.      From line 86: It of course inevitable for this ECFAS project to have multiple associated names, but for the reader this is a bit unclear. What is the difference between ECFAS Pan-EU Impact Catalogue and ECFAS Flood impact layers generated for ECFAS Flood Catalogue. What is the ECFAS CEMS framework?**

The ECFAS Pan-EU Flood Catalogue consists of flood maps covering most of the European coast, describing 15 flood scenarios of maximum TWL and duration for each of the defined coastal sectors (see Le Gal et al. (2023)* for more details). The ECFAS Pan-EU Impact Catalogue collects layers of impacts to population and other assets, such as buildings, roads, etc, and it was produced using the flood maps from the ECFAS Pan-EU Flood Catalogue.

The paragraph was edited on L87-97 to avoid confusion between the different products and the methodology presented in this paper.

*Le Gal, M., Fernández-Montblanc, T., Duo, E., Montes, J., Cabrita, P., Souto Ceccon, P., Gastal, V., Ciavola, P. and Armaroli, C. (2023). A new European coastal flood database for low–medium intensity events. Natural Hazards and Earth System Sciences, 23(11), 3585-3602.

**4.      Section 2.1: On which basis were the extreme weather events and cities selected?**

The extreme coastal events and locations were retrieved from the ECFAS database of extreme events (Souto Ceccon et al., 2021)*. The database contains information of extreme coastal events in the period 2010-2020 that were identified based on information collected through publicly available resources, Copernicus Emergency Management Service activations, and from other flood impact databases. The ECFAS database contains events that generated significant flooding and impacts along EU coastlines, and therefore it was used to retrieve coastal flood impact data necessary to perform the analysis and to build the impact

tool and catalogue of impacts at pan-EU scale. Additional information can also be found at Souto Ceccon et al. (2024)**.

Information about the database used to retrieve the extreme weather events and affected cities, together with a new reference, was added to Section 2.1 (L108-112) in order to clarify this aspect.

*Souto Ceccon, P., Duo, E., Ciavola, P., Fernández Montblanc, T., Armaroli, C., 2021. Database of extreme events, test cases selection and available data, Deliverable 5.1 – ECFAS Project

** Souto-Ceccon, P. E., Montes-Perez, J., Duo, E., Ciavola, P., Fernández Montblanc, T., and Armaroli, C.: A European database of resources on coastal storm impacts, Earth Syst. Sci. Data Discuss. [preprint], https://doi.org/10.5194/essd-2024-183, in review, 2024.

**5.        Section 3.2: The correction of interpolated values using the ratio between the cell areas of the flood map and the datasets is mentioned. It would be beneficial to explain the rationale behind this correction in more detail, as it could be a critical step in ensuring accuracy.**

The number of people affected by coastal flooding was carried out using the Global Human Settlement - Residential Population (GHS-POP) and ENACT layers, with a spatial resolution of 250 m and 1 km respectively. For each cell of the layer, the value represents the absolute number of inhabitants in the cell, and it is therefore dependent on the cell area. Given that the spatial resolution of the flood layers used in this study (100 m) is better than the population datasets, the population layers were interpolated (nearest neighbor) using as reference the center of the cells of the flood maps. The interpolated values were corrected by multiplying them by the ratio between the cell areas of the flood map and the dataset to take into account the different cell resolutions. For example, the ratio between the cell areas of the flood maps and the GHS-POP layer  is 0.16: if the interpolated value is 100 people (in a cell with a resolution od 250 m), the corrected value is 16 people (in a "flooded" cell with a resolution of 100 m). This is reported in the manuscript at L203-209.

The first paragraph of Section 3.2 (L203-213) was improved to clarify this aspect.

**6.        Section 3.2: The section mentions that the datasets were interpolated to match the spatial resolution of the flood model. Please provide more information which interpolation method is used and provide elaboration on why despite which limitations the upscaling has been selected.**

The interpolation used was the nearest neighbor method, which, after several tests, proved to be the most reliable in comparison to the linear interpolation or other common methods. Please, see the previous answer (Comment 5) for details on the correction applied to match the resolution of the flood maps.

**7.      Section 3.3: The manuscript mentions the use of an ensemble approach based on FDCs, citing Figueiredo et al. (2018) and Duo et al. (2020). It would be helpful to briefly explain how this ensemble approach works and its advantages in the context of building damage evaluation.**

The model ensemble approach is a probabilistic-based assessment that relies on the combination of flood damage curves from different impact models. Most impact models are deterministic, but different studies have shown that the use of multi-models produces better results. The result of these types of approaches, like the one from Figueiredo et al. (2018), provides reliable probabilistic damage estimates that are more useful results for interpretation and decision making. For model ensembles, results improve as more models are considered.

Additional information was added to the section of the general aspect (Section 3.1; L186-196) to briefly describe the ensemble approach and how it works.

**8.      Table 5: at Xaver road impact range is not presented.**

This was an oversight from an older version of the manuscript. In the meantime, results were rerun and refined. The current version (Appendix D for details) reports 24.4 – 45 k€/km (32.9 k€/km) as road impact for Xaver. The missing information was added to Table 5.

**9.      Section 5: The manuscript adequately acknowledges the alignment of simulated damage with reported ranges for residential buildings in the Xaver storm, emphasizing the potential for overestimation and attributing it to uncertainties inherent in the probabilistic approach. However, it would greatly benefit from a more detailed elaboration on the specific sources of uncertainty, such as the assumptions made in vulnerability modeling or the variability in reported data. Similarly, while the underestimation of damage for commercial buildings and the mixed category is acknowledged, a deeper exploration into the nuanced factors contributing to this discrepancy and their implications for the model's reliability in diverse urban environments is warranted. Additionally, the discussion on the significant gap between simulated and reported road damage is informative, yet a more thorough analysis of whether the model adequately captures the diverse characteristics of different road types and the potential reasons behind this observed difference would enhance the manuscript's comprehensiveness.**

Validating impact models for large-scale applications is a difficult process due to several factors, such as the limited availability of reliable data and the fact that data are often provided in an aggregated form. In the present work, an extensive effort has been made to collect data on the impacts generated for different reference cases along the European coasts, performing the validation with impacts generated by 3 historical events that impacted coastal areas with different characteristics.

In this type of analysis, the selected flood model, the flood damage curves and the reported damage can introduce uncertainty into the study. An under- or overestimation of the flood extension or of the flood depth could lead to an under- or overestimation of the damage. In Section 2.2 the characteristic of the used flood maps is explained. In the case of the flood damage curves, this paper uses an ensemble approach. For example for commercial

buildings, 4 different flood damage curves were used to build the ECFAS Impact Model. These flood damage curves may have discrepancies at certain values of flood depth. In section 3.3 the used models are described. Finally, in section 3.5 (table 4) an analysis of the reliability and the representativeness of the different resources used for the validation of the modelled results is presented; an analysis of the confidence of the data used for validation is presented in section 5.1.4, and a description of the limitations of the methodology is presented in section 5.3.

**Response to the reviewer 2**

Dear referee, many thanks for your time and feedback. This reply contains the detailed answers to the comments provided in the first reply, modified where necessary according to the changes made in the text, which is also attached to show the changes. A reference to the part of the text that has been modified as a result of the comments is also included for each comment.

**1.      Conclusions: The concluding remarks read more like an endorsement of a specific EU project and prospects for future model advancements rather than a reflection on the key findings of the paper itself. It's essential to rephrase them to emphasize the main takeaways from the study. What insights were gained? Avoid introducing new elements; those can be addressed in the discussion section.**

The conclusion section (L549-571) was reviewed and edited to better highlight the key findings of the paper.

**2.      Validation: The comparison between computed damages and observed damages in the three reference cases reveals disparities, which is understandable given the incomplete input and damage data. While the authors attempt to account for these differences, the explanations seem more like attempts to justify them. It's crucial to maintain a factual approach.**

The scarcity of reliable data on the impact of storms, usually available only in aggregated form, makes validation of impact models for coastal areas difficult. In this study, a validation process was carried out for 3 historical events that impacted different coastal areas (Section 5.1), together with an analysis of the reliability and representativeness of the different resources used for the validation of the model output (Section 5.1.4). Although the authors agree that an analytical approach would be more appropriate for comparison with the results, due to the type and amount of available data an expert judgment approach was considered appropriate.

**3.      Scope of Analysis: Given the discrepancies observed in the three reference cases and the explanations needed to analyze these, I advise not presenting the additional sites. What additional value does it offer? The results for these cases are inadequately presented and explained (only in the context of grid-based vs object-based). I strongly advise to focus solely on the three reference cases in this paper. There is adequate scope to write a paper on this.**

Although the authors agree that the information for the 3 reference cases provide key information for the scope of this study, the test cases represent important additional information because they have been used to compare the object-based methodology presented in the study with a grid-based analysis that is widely used in similar studies by the scientific community, therefore we think that they represent a supplementary useful information for the reader. In fact, the test cases represent sites that have experienced coastal storms that generated remarkable impacts in the period 2010-2020. The validation of these results is not possible due to the lack of data and because the direct comparison between the object-based and the grid-based approaches is not meaningful.

**4.        Clarity on Probabilistic Approaches: The frequent use of the term "probabilistic" lacks immediate clarity regarding the specific stochastic processes employed. It would be beneficial to provide a concise overview of the stochastic methodologies used. Which stochastic variables do you consider? Which not, and why not?**

Additional information regarding the model ensemble approach and its ability to provide an uncertainty estimation was added to section 3.1 to clarify the approach(See Answer to Reviewer 1, Comment 7).

The probabilistic resample was applied to the number of people affected by the flood, to the financial damage to buildings and to roads. Given the different approaches, the resampling was different for each damage sector. The general description, common to population, buildings and roads, was described in Section 3.1, while the specific information was detailed in sections 3.2, 3.3 and 3.4. Some clarifications are presented below:

- Population (number of affected people): the method generates an ECDF based on a set of values of the number of affected people in each flooded cell. The set is built using multiple sources of population density. Then, for each flooded cell, a large number of values (1000) is randomly generated from the ECDF, representing the probabilistic estimation (distribution) of the number of affected people in that flooded cell. The process is applied to each flooded cell and the probabilistic estimate of the total affected population is based on the resampling set of each cell. This process is described in L203-213 and in Figure 2.
- Buildings (relative damage): the method generates an ECDF based on a set of values of relative damage factor for each flooded building. The set is built using a set of damage curves. Then, for each flooded building, a large number of values (1000) is randomly generated from the ECDF, representing the probabilistic estimation (distribution) of the damage factor for that flooded building. The distribution is then multiplied for the (deterministic) maximum damage to retrieve the financial damage. The process is applied to each flooded building and the probabilistic estimate of the total building damage is based on the resampling set of each building. Additional information was added to section 3.3 to clarify this aspect.
- Roads (relative damage and maximum damage): the probabilistic resampling is applied to both the relative damage (using multiple damage curves) and the maximum damage (using an empirical set). As before, in both cases the method generates an ECDF based on a set of values of relative damage factor for each flooded road, and a set of values of maximum damage, and the ECDFs are resampled (n=1000). The resampling of the relative and maximum damage are combined generating a set of n x n values of financial damage for each road that represent the probabilistic estimation. This is described in Section 3.4 and in Figure 5. The caption of Figure 5 was modified to include some of the information therein in the main text.

Note that other variables were not chosen for the probabilistic estimates mainly because of the lack of available data or the lack of multiple models for the implementation of the model ensembles.

Information about the general description was added to Section 3.1 (L190-196). Specific information about buildings was added to Section 3.3 (L249-253). For roads, the caption of Fig.5 was modified (L285-289) and a sentence was added (L275).

**5.      Treatment of Population Data: In one instance, it appears that population data is treated as a stochastic variable. However, given the deterministic nature of population demographics, this approach seems unwarranted. You know the month and time that an event impacted a particular coast. It is more appropriate to explore the sensitivity of the results to different population datasets by comparing the two data sets (thus one of the 24 elements of the second data set).**

The number of people affected by the coastal flood is evaluated by both the Global Human Settlement - Residential population (GHS-POP) and ENACT datasets. There are different sources of uncertainty from the selected datasets: related to the temporal reference of the datasets (ENACT: 2011; GHS: 2015) in comparison with the date of the flood event, and related to the differences between flood maps and the spatial resolution of the datasets. The seasonal and night/day variability is accounted for by applying the probabilistic resampling of the datasets. At the operational level, an evaluation of the affected population based on the timing of the coastal extreme is expected to be more appropriate. Certainly, it would represent a refinement of the assessment, from a deterministic point of view. By evaluating the number of people on the test case AoIs, the results of the comparisons between the GHS-POP and the yearly average night/day and summer/winter seasons showed minor variability in terms of magnitude of the people's presence. In general, all evaluations identify a similar number of affected people, if variations within the same magnitude are considered acceptable. For these reasons, a probabilistic implementation was preferred using the combination of the different datasets.

**6.      Sensitivity Analysis of Vulnerability Curves: While the vulnerability curves significantly impact the results, they are not inherently probabilistic. Reframing the discussion around sensitivity analysis might be more accurate.**

The authors agree that an FDC is not a probabilistic model itself. However, using multiple curves as multi-model ensembles is recognized to generate probabilistic estimates, as long as the result is represented as a distribution (mean-dev.stand, quantiles, etc…) generated by combining the curves. Information was added to Section 3.1 to clarify this and other issues highlighted by the reviewers (L186-196). Furthermore, a statement related to this comment was added in the conclusions (L563-564).

**7.      Wind: how is damage due to wind treated and isolated in the damage reports?**

The effect of wind is not isolated from the reported damage, so the reported damage was interpreted with caution. Although the authors would like to have disaggregated data, it is in most cases impossible to access it and it is a limitation of this type of studies. Nevertheless, the selected reference cases have been thoroughly analysed, and although wind can play an important role in the damage, the impacts generated by the selected events have mainly been caused by flooding.

**8. Estimation of Cost Data: The utilization of probabilistic methods for estimating cost data requires clarification on the approach employed.**

A brief description of the ECDF-based approach of cost estimation is present in the manuscript at L270-278, which includes: references to the dataset used from Van Ginkel et al. (2021), reference to the details in the Appendixes, and a figure (Figure 5) that shows the representativeness of the ECDFs applied in comparison to the original dataset. Additional information about the ECDF observed and the ECDF calculated was added (L275).

**9. Assessment of Flood Maps: It seems that the flood maps are considered as given, obtained from an external source. It would be valuable to evaluate the accuracy of these maps, particularly in the context of the three reference cases. What was the bias between the model results and the observations in terms of high water marks and flood extends?**

The flood maps were retrieved from the European flood catalogue implemented in the ECFAS project and described in the paper by Le Gal et al. (2023)\*. The process of flood map generation and validation can be found in Le Gal et al. (2023)\*. The calibration and validation of the numerical model used for the generation of the flood maps (LISFLOOD-FP) can be found in Le Gal et al. (2022) and Le Gal et al. (2024)\*\*\*.

\*Le Gal, M., Fernández-Montblanc, T., Duo, E., Montes, J., Cabrita, P., Souto Ceccon, P., Gastal, V., Ciavola, P. and Armaroli, C. (2023). A new European coastal flood database for low–medium intensity events. Natural Hazards and Earth System Sciences, 23(11), 3585-3602.

\*\*Le Gal, M., Ciavola, P., Gastal, V., Fernández-Montblanc, T. and Delbour, S. (2022). Validated LISFLOOD-FP model for coastal areas, Deliverable 5.2 – ECFAS Project (GA 101004211), www.ecfas.eu (Versión 2). Zenodo. https://doi.org/10.5281/zenodo.7488694

\*\*\*Le Gal, M., Fernández-Montblanc, T., Montes, J., Souto Ceccon, P., Duo, E. and Ciavola, P.; Influence of model configuration for coastal flooding across Europe, Coastal Engineering, 104541, https://doi.org/10.1016/j.coastaleng.2024.104541, 2024

**10. Explanation of Methodology: The paper contains numerous references to methods described in other works, making it challenging for readers not familiar with the referenced EU project to follow. Providing more detailed explanations of the methodologies employed would enhance comprehension.**

In this paper several damage models from previous studies are used to build the ECFAS impact approach based on a multi-model ensemble. A brief description on the selected FDCs to calculate the impacts on buildings along with an indication of the macro-class for which a curve was available (Residential, Commerce and Industry) can be found in Table 3 and Figure 3. For roads, based on Van Ginkel et al. (2021), a detailed description of the FDCs, the characteristics of roads and hydrodynamic flow conditions, number of lanes and construction cost ranges is provided in Appendix C.

Additionally, the text has been improved by adding relevant information for some key references (L242-243; L262-263), i.e., Huizinga et al. (2017) and Van Ginkel et al. (2021).

**11.    Reference Clarification: There is a discrepancy where section 3.1 in Line 230 is referenced, but no description of the probabilistic evaluation is provided therein.**

In section "3.1 General aspects", paragraph 2 (lines 185-196), reference is made to the probabilistic evaluation. As this aspect was common for population, buildings and roads, the authors decided to add it in a general section to avoid repetitions.

**12.    Inclusion of Damage Maps: Please show maps depicting computed and observed damages . Incorporating such visual representations, including building footprints and damage extents, would provide valuable context for interpreting the aggregated results.**

An appendix with modelled impact maps for the reference cases was added (Appendix E) and referred to in the text (L328). The maps contain information on the floods used to model the impacts, the estimation of the affected people, a damage assessment for buildings (number of flooded buildings and mean damage) and a damage assessment for roads (affected road length and mean damage). Even if the scale of the maps does not allow to depict all details (e.g., impacted buildings), the information on impacts and flood extent is also provided in the legend. The figures provide an overview of the impacts at the reference case scale.

**13.    Line 266: What is a semi-quantitative, holistic comparison?**

The terms were deleted to avoid adding unnecessary long explanations (Line 291).

**14.    Figure 8: the uncertainties seem very small. Is this because the variations in Figure 10 are so small?**

The small uncertainty for affected population values are due to the fact that the used datasets (ENACT and GHS-POP) provide similar information, as shown in Fig.10 and discussed in Section 5.3.1. To note that in both figures (Figure 8 and 10) the representation of affected people follows a logarithmic scale.

**15.    Table 5 Xaver/Roads still has xx's.**

This was an oversight from an older version of the manuscript. In the meantime, results were rerun and refined. The current version (Appendix D for details) reports 24.4 – 45 k€/km (32.9 k€/km) as road impact for Xaver. The missing information was added to Table 5.

[revised manuscript text omitted]

---

## Author Response (AR2)

Juan Montes

Earth Sciences Department, INMAR, University of Cadiz

25/07/2024

Dear Editor Robert Sakic Trogrlic,

We thank you for your time and for the opportunity to submit the reviewed version of the manuscript entitled "Validated probabilistic approach to estimate flood direct impacts on the population and assets on European coastlines".

This document contains both the detailed responses to the reviewer's comments and the manuscript with tracked changes to show the modifications made in the review. In addition, a reference to the part of the text that has been modified as a result of the comments is included in the replies. The lines' numbers refer to the new track-changes manuscript version.

We hope that the modifications made to the manuscript and the answers provided to the reviewer will clarify the doubts raised during the review and improve the document to be considered for acceptance.

Many thanks,

Juan Montes

**Response to the reviewer**

Dear reviewer,

Many thanks for your time and feedback, which we consider to be significantly improving the manuscript. We have tried to incorporate all the requested information in the manuscript, and not just in the rebuttal table.

This reply contains the detailed answers to the comments, and the manuscript with tracked changes is also attached to show the modifications. Previous revisions are not shown in the new version of the manuscript. A reference to the part of the text that has been modified is also included for each comment.

**Dear Editor,**

**Unfortunately, the authors did not address my review comments sufficiently. In some cases some changes were made to the text but in other cases only a (lengthy) rebuttal was provided without any changes in the manuscript. So I will re-iterated the points in brief:**

**1. Conclusions. Changes were made to make it read less like an ECFAS marketing brochure but now it is more summary than conclusions. What are the main take homes from this study, other than that it is now "better"?**

In addition to the changes made to the conclusions to make them less focused on the ECFAS project, the section was reviewed and edited to better highlight the key findings of the paper (L576-611). The conclusions have been extensively modified to emphasise methodological innovations, validation and reliability, the scale, the limitations, the practical applications and future research.

**2. Validation. The arguments given in the rebuttal are valid, but they are not stated in the manuscript. Please add these considerations so future readers will understand these difficulties.**

The information given in the rebuttal table was added to Section 5.1.4 (Confidence), which is part of the section about the validation with reported impacts (Section 5.1). The modifications, applied in L481-486, have highlighted the complexity of validating the impact models due to the scarcity of reliable data, the effort made to do a validation in three different reference cases and the fact that the comparison was based on an expert judgement approach.

**3. Scope. If the additional cases are so important for this study, which is so little text spent on Figure 8? If you keep it in (which I advise not to) then you have to discuss the results and findings in more detail.**

The additional test cases were used to show the applicability of the ECFAS Impact Approach on a European scale. They represent well-known storm events in the period 2010-2020 that have generated major impacts on the AoIs. Furthermore, the results have been used to make

a comparison between the object-based and probabilistic ECFAS Impact approach and the widely used grid-based analysis. The grid-based analysis has been frequently used in impact assessments at large scales, therefore the authors considered it essential to compare the two methods and illustrate the improvements that the ECFAS Impact approach can provide.

Additional information was added in the text to improve this section (L364-373) and to complement the information included in Table D 1 in Appendix D that gives an overview of impacts on population, buildings and roads for the test cases.

**4. Clarity. There is a lot of information provided to the reviewers in the rebuttal that would be of interest to future readers. Please use these texts in the manuscript.**

In addition to the information added in the General aspect section (3.1), in Buildings section (3.3) and in the Roads section (3.4) of the methodology, additional information from the rebuttal table was added in the text (Section 3.4, L279-283).

**5. Population data: I have the impression that the authors missed my point that you cannot use a dataset for population variations over the calendar months and day/night as a proxy for the potential variation of a population during the storm season. I re-read the rebuttal text several times but I cannot make sense of it.**

Authors agree that an evaluation of the affected population based on the timing of the coastal storm is expected to be more appropriate and would represent a step forward for the assessment of the ECFAS Impact approach. The ENACT database could be used for this purpose as this spatial raster dataset depicts the distribution/density of population during night-time and daytime for each month. However, an analysis of the day/night and monthly variations on the test case AoIs showed a low variability of the population density. Variations in population density were also analysed by differentiating the summer (April-September) and winter (October-March) periods, which also showed low variability. Comparing all these results also with the static layer GHS-POP, that give the distribution of persons in 2015, minor differences were observed, being remarkable only in the Castellon test case. All these results are shown in figure 10 of the manuscript. In addition, the databases used, the best available for this analysis at European level, are from 2011 and 2015, which implies that the population distribution for recent events is not available on a pan-European level.

For the aforementioned reasons, a probabilistic implementation was preferred using the combination of the different datasets instead of using only one database to retrieve the population density for a specific event. The approach could be improved if more detailed information on the seasonal/day-night variation of the population will be available.

**7. Wind: Again, great answer in the rebuttal, but add a statement about this in the manuscript. This is of interest to future readers.**

Information regarding the wind effect on damage has been incorporated into the manuscript (L486-489). The additional text clarifies that the impact of wind is not isolated from the reported

damage due to the lack of disaggregated data. Additionally, the modified text emphasized that the selected reference cases have been thoroughly analyzed to ensure that the impacts were predominantly caused by floods.

**9. Flood maps. Great references, but what I was looking for was a statement in your manuscript about the accuracy of these maps for the case studies, in relation to all the other unknowns you are dealing with.**

Additional information regarding the methodology used to validate the flood maps and the accuracy of these maps has been added to L138-141. Readers are also directed to the original publications from which the flood maps were obtained for more details. Further information on the validation results, as well as specifics about the model and data used, can be found in the cited references.

Additionally, details about the accuracy of the flood maps for the three reference cases and their implications for the validation process have been included in the discussion section (L490-494). This new section highlights that the accuracy of the modelled flood extent can bias the validation process, potentially leading to under/over-estimation of impacts.

**12. Damage maps: ok. The word "water thickness" is strange. I would use "water depth"**

The word "Water Thickness" was replaced by "Water depth" in the legend of the figures in Appendix E.

**14. Ok, but again write about this in the manuscript.**

A sentence was included in the manuscript (L519-520) addressing the small uncertainties in affected population values, as the datasets used, ENACT and GHS-POP, provide similar information. This addition is also related to comment 5 and supports the decision to use a probabilistic approach for estimating the affected population, rather than a deterministic value based on the impact storm date. The scientific literature states that uncertainties due to the input datasets are expected for this type of assessment, however the probabilistic resampling partially mitigates it.

Additionally, a statement indicating that the y-axis follows a logarithmic scale was added to the caption of Figure 8 (L375-776).

**Additional comment: Line 186: " The ensemble approach usually works better than deterministic ones". On what basis is this statement made?**

The statement that model ensemble approach usually works better than deterministic ones is widely established in the literature, not only for flood damage curves, but also in the use of other types of models as for instance stated by Montaño et al. (2020)* and Senechal and Coco

(2024)** for shoreline evolution models or as established by Garzon et al., (2024)*** for coastal erosion. In this case, the statement directly refers to Figueredo et al. (2018) paper, as this work is closely related to the objective of this paper.

The sentence was slightly modified to clearly state that the reader can refer toFigueredo et al. (2018) (L187).

*Montaño, J., Coco, G., Antolínez, J.A.A. et al. Blind testing of shoreline evolution models. Sci Rep 10, 2137 (2020). https://doi.org/10.1038/s41598-020-59018-y

**Coco, G. On the Role of Hydrodynamic and Morphologic Variables on Neural Network Prediction of Shoreline Dynamics. https://doi.org/10.1016/j.geomorph.2024.109084

***Garzon, J. L., Ferreira, O., Plomaritis, T. A., Zózimo, A. C., Fortes, C. J. E. M., & Pinheiro, L. V. (2024). Development of a Bayesian network-based early warning system for storm-driven coastal erosion. Coastal Engineering, 189, 104460. https://doi.org/10.1016/j.coastaleng.2024.104460

[revised manuscript text omitted]